# Highly efficient platelet generation in lung vasculature reproduced by microfluidics

Xiaojuan Zhao [1] ✉, Dominic Alibhai [2], Tony G. Walsh [1], Nathalie Tarassova[1], Maximilian Englert [3], Semra Z. Birol[1], Yong Li [1], Christopher M. Williams[1], Chris R. Neal[2], Philipp Burkard [3], Stephen J. Cross [2], Elizabeth W. Aitken[1], Amie K. Waller[4], José Ballester Beltrán [4], Peter W. Gunning [5], Edna C. Hardeman [5], Ejaife O. Agbani [6], Bernhard Nieswandt [3], Ingeborg Hers[1], Cedric Ghevaert[4] & Alastair W. Poole [1] ✉

Platelets, small hemostatic blood cells, are derived from megakaryocytes. Both bone marrow and lung are principal sites of thrombopoiesis although underlying mechanisms remain unclear. Outside the body, however, our ability to generate large number of functional platelets is poor. Here we show that perfusion of megakaryocytes ex vivo through the mouse lung vasculature generates substantial platelet numbers, up to 3000 per megakaryocyte. Despite their large size, megakaryocytes are able repeatedly to passage through the lung vasculature, leading to enucleation and subsequent platelet generation intravascularly. Using ex vivo lung and an in vitro microfluidic chamber we determine how oxygenation, ventilation, healthy pulmonary endothelium and the microvascular structure support thrombopoiesis. We also show a critical role for the actin regulator Tropomyosin 4 in the final steps of platelet formation in lung vasculature. This work reveals the mechanisms of thrombopoiesis in lung vasculature and informs approaches to large-scale generation of platelets.

Platelets are small anucleate blood cells[1,2], with critical roles in hemostasis, thrombosis, inflammation, vascularization, innate immunity and tissue regeneration[3,4]. Platelets are formed from mature polyploid megakaryocytes (MKs), their precursor cells[5], although the process of their generation remains incompletely understood[6,7]. In vitro studies suggest that dynamic changes[8] and regulation[9] in the actin cytoskeleton play important roles in platelet production from MKs.

Bone marrow is proposed to be the main site of MK maturation and platelet production, however much of the evidence since the original observations in 1893 are indirect. MKs have been observed within lung tissue or vasculature[10], and more platelets and fewer MKs have been shown to be in the blood exiting the lungs compared to blood entering the lungs[11,12], supporting the concept that platelet generation may take place from circulating MKs in the lung vasculature. Recently, direct evidence presented by Lefrancais et al. has shown that the lung is a primary site of platelet biogenesis[13]. Data from the Poncz group also show that intravenously infused murine[14] or human MKs[15] release functional platelets within the lungs of recipient mice. However, the mechanisms underlying platelet biogenesis in the lung vasculature have not been explored.

In vitro-derived platelets, as an alternative to native platelets, are attractive for fundamental research because of their rapid genetic

[1]School of Physiology, Pharmacology and Neuroscience, Biomedical Sciences Building, University of Bristol, Bristol BS8 1TD, UK. [2]Wolfson BioimagingFacility, Biomedical Sciences Building, University of Bristol, Bristol BS8 1TD, UK. [3]University Hospital and Rudolf Virchow Center, University of Würzburg, Würzburg D-97080, Germany. [4]University of Cambridge / NHS Blood and Transplant, Wellcome-MRC Cambridge Stem Cell Institute, Jeffrey Cheah Biomedical Centre, Cambridge Biomedical Campus, University of Cambridge, Cambridge CB2 0AW, UK. [5]School of Medical Sciences, University of New South Wales, Sydney, NSW 2052, Australia. [6]Cumming School of Medicine, University of Calgary, Calgary, AB T2N 1N4, Canada. ✉e-mail: xz14926@bristol.ac.uk; A.Poole@bristol.ac.uk

tractability, as vectors for drug and genetic component delivery[16] and in clinical platelet transfusion. At present, however, the inability to generate large number of functional platelets efficiently in vitro is a major obstacle.

In this work, we established an ex vivo mouse heart-lung model (Fig. 1a) through which we were able to perfuse murine MKs. Remarkably, we could show for the first time that MKs, despite their large size (50–100 μm)[5], can pass multiple times through the lung vasculature, and that this leads to the generation of very large number of fully functional platelets (up to 3000 per MK[6,17]). Using this system and an in vitro microfluidic chamber we show roles for oxygenation, ventilation, pulmonary endothelium and the micro-vascular structure in platelet generation, demonstrating how the lung may be uniquely suited to thrombopoiesis. Our data show that MKs undergo enucleation upon repeated passage through the pul-monary vasculature prior to platelet generation, with the final steps dependent on the actin regulator Tropomyosin 4 (TPM4). This contrasts with our observations of MK morphologies in the pro-gression of MKs from marrow space to sinusoid, which looks similar to wild-type (WT) controls, suggesting distinct mechanisms in the final steps of thrombopoiesis in the lung vasculature versus the bone marrow. Altogether, our study advances our understanding of pla-telet formation and establishes an approach to generate large number of them outside the body.

## Results

### Efficient platelet generation in lung vasculature ex vivo

The lung has been proposed as a site of platelet generation by several groups periodically, there is now a need to understand the mechanism underlying this platelet generation. For this reason, we established an ex vivo mouse heart-lung model (Fig. 1a) through which we were able to perfuse murine MKs. The ex vivo mouse heart-lung model is based upon isolating the heart and lungs as a single unit, ligating the venae cavae and the aortic arch and perfusing prestained mature MKs (Supplementary Fig. 1a shows the experimental flowchart, Supple-mentary Fig. 1b shows DNA ploidy analysis for cultured MKs, Supple-mentary Fig. 1c shows the diameters of perfused MKs, Supplementary Fig. 1d shows staining of demarcation membrane system (DMS) with PE- or FITC- conjugated CD41 and nuclei with Hoechst 33342, respec-tively) through the pulmonary circulation from the right ventricle, collecting the perfusate from the left ventricle. This allows quantita-tion and imaging of cells passaged through the pulmonary vasculature. In the first instance, lungs were artificially ventilated with air. We were expecting that the vast majority of MKs would be trapped within the lung vasculature due to their large size (around 50–100 μm diameter)[5], but unexpectedly more than 50% of the intact MKs (showing a circular shape and central nucleus) emerged in the perfusate after the first passage (Fig. 1b). The perfusate could be re-injected through the lungs and upon multiple passages, the numbers of intact MKs in the perfu-sate continued to decrease (Fig. 1b). We also demonstrated that intact mouse MKs could pass through the pulmonary vasculature in vivo, as intact mouse MKs prestained with CellTracker™ Red CMTPX dye and Hoechst 33342 could appear in the blood of the left common carotid artery of an anaesthetized recipient C57BL/6 after being infused into the right external jugular vein (Fig. 1c). It was also apparent that CD41-labelled particles also appeared in the perfusate that were similar in size and granularity to mouse platelets. We term these "generated" platelets (CD41-positive events in gate P1 in Fig. 1d–e). The generated platelets were bona fide live platelets rather than cellular fragments, using the vital dye Calcein-AM (Fig. 1e) and showing mitochondrial membrane potential comparable to normal mouse platelets using tetramethyl rhodamine methyl ester (TMRM, an indicator of healthy cells, determined by accumulation of TMRM in active mitochondria) (Fig. 1f–g). Generated platelets were anuclear, showing no staining with the DNA dye Draq5 (Fig. 1e). Supplementary Fig 1e shows

DRAQ5 staining for MKs, which provides a positive control for this stain. The numbers of generated platelets per MK gradually increased with increasing passages, up to around $931.7 \pm 138.4$ platelets/MK after 18 passages (Fig. 2a, c). Two-photon microscopy of fixed lung sections after 18 passages showed that many generated platelets could be seen in the lung microvasculature (Fig. 2b and Supplementary Movies 1–3). These were quantified in a defined lung volume and the total lung volume measured using a fluid displacement method (Supplementary Fig. 1f) to estimate numbers of retained platelets. We calculated this to be $2066.0 \pm 274.7$ per MK injected (Fig. 2c). Adding this to the number contained in the perfusate (Fig. 2a, c), we estimate that after 18 pas-sages through the lung ~$2997.0 \pm 270.5$ platelets were generated per MK (Fig. 2c), in keeping with previous estimates that each MK in vivo produces ~1000–4000 platelets[6,17]. Platelets were not generated sim-ply as a consequence of passage through small-bore needles, since we showed that repeated passage of MKs through 21 G needles 18 times is not sufficient to generate platelets (Fig. 2c). Altogether, we could generate physiological numbers of platelets after multiple recircula-tion of MKs through the lung microvasculature.

### Mechanisms of platelet generation in mouse vasculature

The ex vivo mouse heart-lung model (Fig. 1a) can be a useful tool to allow artificial ventilation with either ambient air or with pure nitrogen, or no ventilation, to assess the roles of physical ventilation and gas-eous oxygen in regulating MK biology and thrombogenesis. We first explored whether air ventilation is essential for platelet generation in our model. In the absence of ventilation, the numbers of platelets generated per MK in the perfusate still gradually increased with increasing passages ($422.6 \pm 118.7$ platelets/MK, Fig. 2a), but the num-bers generated were substantially lower than in the air-ventilated condition. Two-photon microscopy of fixed lung sections after 18 passages showed that fewer generated platelets could be seen in the lung microvasculature (Fig. 2b and Supplementary Movie 4), com-pared to the air-ventilated lung. Therefore this indicates that air ven-tilation is important in platelet generation in the lung, but may result either from an effect directly on MKs and/or through an effect on pulmonary endothelium. Pulmonary endothelial cells (ECs), which play key roles in gas exchange in the lung[18], interact closely with MKs as they passage through the vasculature. We therefore compared their viabilities (determined by the Calcein Deep Red retention assay) and mitochondrial membrane potential in the lungs under air ventilation or unventilated conditions for ~2 h. Surprisingly, ECs from prepara-tions of unventilated lungs were fully viable, and comparable with those ventilated under air (Fig. 3a). However, the mean intensity of TMRM of ECs from unventilated lungs was approximately half that of air-ventilated lung (Fig. 3b).

Strikingly, when lungs are ventilated with pure nitrogen to com-pletely de-oxygenate the heart-lung preparation, the number of gen-erated platelets in the perfusate was almost ablated, reduced to just $43.2 \pm 16.7$ platelets/MK after 18 passages (Fig. 2a). Two-photon ima-ging of nitrogen-ventilated lungs showed mature MKs were trapped in the lung vasculature (Fig. 2b and Supplementary Movie 5), a feature not observed under air ventilation or unventilated lung. Importantly, the mean intensity of ECs TMRM was approximately halved by nitrogen-ventilation relative to air-ventilated controls, and equivalent to the unventilated lung (Fig. 3b).

To further verify whether the lung capillary bed could mediate platelet generation, we designed a polydimethylsiloxane (PDMS)-based (gas permeable) microfluidic chamber with channel design simulating tissue microcirculation[19]. The channels were of uniform depth of 10 μm throughout, where the entry and exit channels had a width of 100 μm and where branches emerged halving the channel width each time, to a minimal width of 12.5 μm, as per the diagram shown in Fig. 3c. This channel allowed us to flow through cells and determine platelet generation in the perfusate after repeated

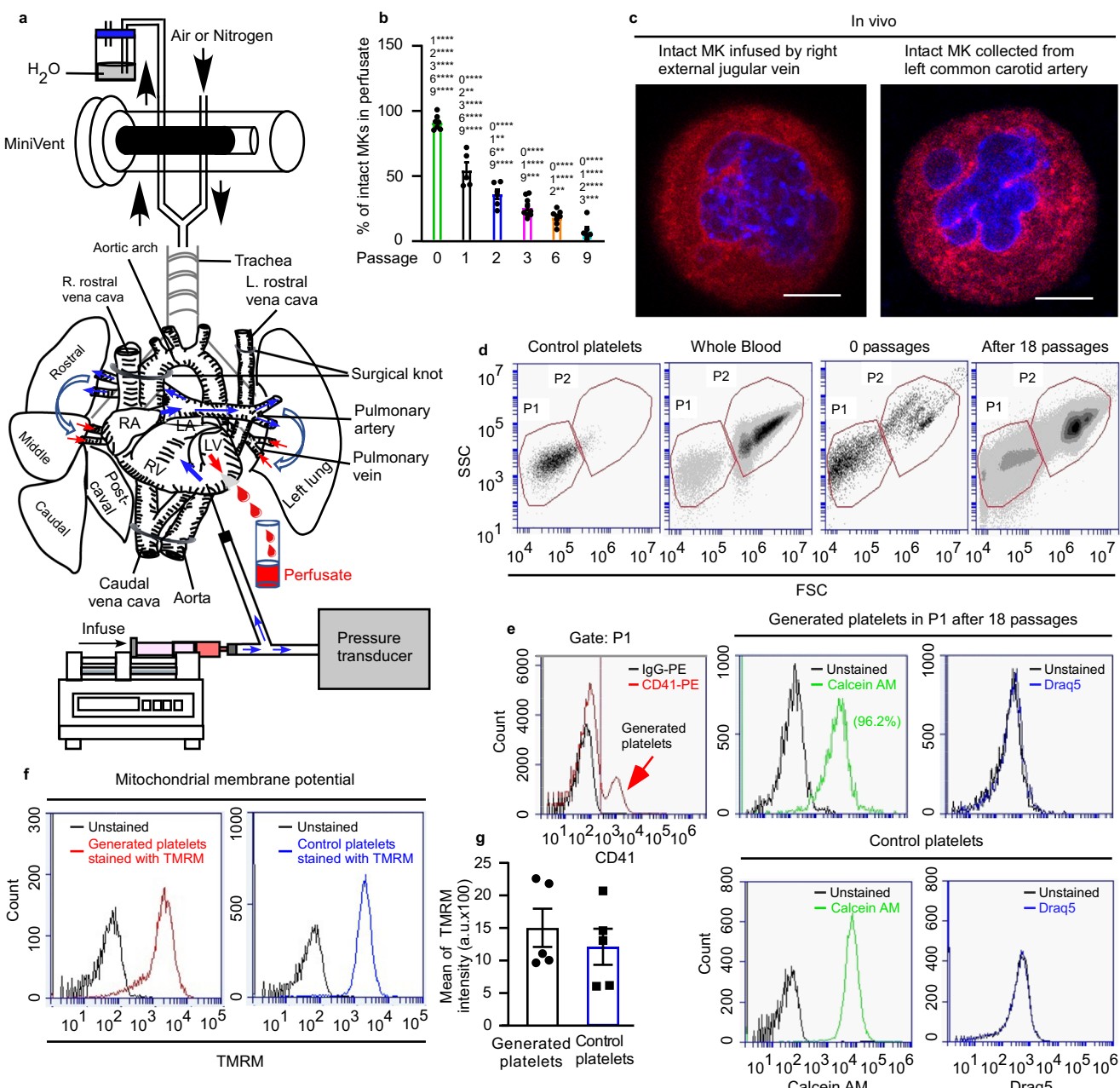

**Fig. 1 | Mouse platelets are generated from megakaryocytes passaged multiple times through mouse pulmonary vasculature ex vivo.** Mouse megakaryocytes (MKs), labelled with CD41-PE or CD41-FITC antibodies, were passaged repeatedly through the pulmonary vasculature ex vivo. Lungs were ventilated with air throughout (**b**, **d**–**g**). **a** Diagram illustrating the approach to generating mouse platelets. End-expiratory positive pressure was applied to prevent lung collapse. **b** Intact MKs from perfusates after passaging the indicated number of times through lung vasculature. Quantification was from ≥250 fields of view, counting ≥230 cells in total and displayed as a percentage of total number of cells. Numbers above each column indicate significant differences to other passages. $N = 7, 5, 6, 9,$ 8 and 6 independent experiments for passage numbers 0, 1, 2, 3, 6 & 9. Error bars are mean ± SEM. Two-way ANOVA with Tukey's multiple comparisons test, $*p < 0.05$, $**p < 0.01$, $***p < 0.001$ and $****p < 0.0001$. **c** In vivo demonstration that intact mouse MKs pass through the pulmonary vasculature. Mouse MKs were stained with CellTracker™ Red CMTPX dye (red) and Hoechst 33342 (blue) prior to injection into the right external jugular vein of an anaesthetized recipient C57BL/6 mouse. Blood was collected from the left common carotid artery and cells were imaged by confocal fluorescence microscopy. Images shown are representative of $n = 4$ independent experiments. Scale bar: 10 μm. **d** Gating strategy for quantification of generated platelets. The number of generated platelets in the perfusate collected after the 18th passage was determined by the number of CD41(+) events in gate P1. **e** Events in P1 gate (from the experiment shown in Fig. 1d) are defined as generated platelets (indicated by the red arrow), with higher mean fluorescence compared to those derived from control IgG-PE-treated MKs. Gate P1 also captures CD41-negative cells, which include stem cells and host-derived platelets. Viability of generated platelets, and whether they contain DNA, were checked by Calcein-AM and DRAQ5 dyes, respectively. **f** Mitochondrial membrane potential in generated and control platelets was determined by Tetramethyl rhodamine methyl ester (TMRM) accumulation in active mitochondria and measured by FACS. **g** TMRM signals from (**f**) were quantified and displayed as mean ± SEM. $N = 5$ independent experiments; two-tailed unpaired $t$ test. Source data are provided in the Source Data file.

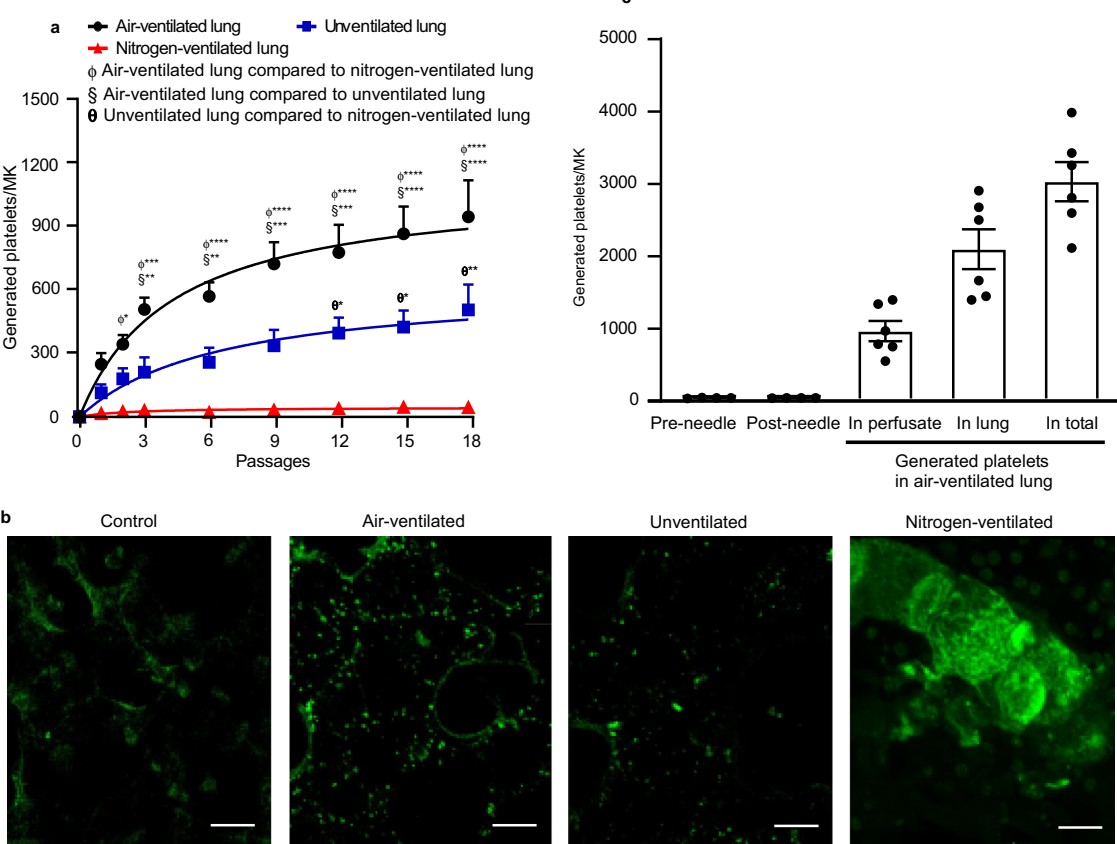

**Fig. 2 | Quantification of platelets generated by passage of megakaryocytes through mouse pulmonary vasculature ex vivo.** Mouse megakaryocytes (MKs), labelled with CD41-PE or CD41-FITC antibodies, were passaged repeatedly through the pulmonary vasculature ex vivo. Lungs were ventilated with air, pure nitrogen or without ventilation. **a** The number of generated platelets per MK present in the perfusates from different passage numbers in lungs either ventilated with air (black circles), pure nitrogen (red triangles), or without ventilation (blue squares) were measured by FACS. $N = 5$ (air-ventilated lung or unventilated lung) and 4 (nitrogen-ventilated lung) independent experiments. Data are mean ± SEM. Two-way ANOVA with Tukey's multiple comparisons test, $*p < 0.05$, $**p < 0.01$, $***p < 0.001$ and $****p < 0.0001$. **b** Stained MKs (CD41-FITC, green) were passaged through

pulmonary vasculature ex vivo 18 times, and lung tissue was fixed and sliced followed by visualization of 20 stacked focal planes by two-photon microscopy. Lungs were either ventilated with air or pure nitrogen, or were not ventilated, as indicated. Mouse lung without MKs passed through served as control. Images shown are representative of $n \geq 4$ independent experiments. scale bar: 20 μm. **c** Numbers of platelets generated per MK in perfusate and retained in mouse lung under air ventilation, were calculated and displayed as mean ± SEM. As a control, MKs passed through 21 G needles 18 times generated no platelets. $N = 6$ (air-ventilated lung) and 4 (passing through needles) independent experiments. Source data are provided in the Source Data file.

passage. Figure 3d−e shows that the numbers of generated platelets, when MKs are flowed through the microfluidic chamber conditioned in normal air, gradually increased with increasing passages, similar to the numbers generated in the unventilated lung, with 492.3 ± 47.6 platelets/MK after 18 passages (Fig. 3e). The generated platelets were live anuclear platelets (Fig. 3d) and showed substantial responses to agonists (thrombin and collagen-related peptide (CRP-XL)) in terms of integrin αIIbβ3 activation and degranulation (P-selectin expression) (Fig. 3f). We also conditioned microfluidic chambers with pure nitrogen to completely de-oxygenate them, causing the generation of platelets to be almost ablated, reducing them to 56.4 ± 1.4 platelets/MK after 18 passages (Fig. 3e), similar to those generated in lungs ventilated with pure nitrogen (Fig. 2a). Altogether, these data suggested that (1) air-ventilation and ECs with normal mitochondrial membrane potential (termed as healthy EC) are required for MKs to generate physiological levels of platelets in the heart-lung preparation; (2) the pulmonary microcirculation plays a role in platelet generation; (3) lack of ventilation or nitrogen-ventilation for 2 h caused partial loss of the mitochondrial membrane potential in pulmonary ECs; (4) exclusion of oxygen from either the lung-heart system or the microfluidic system ablates platelet generation.

## Generated platelets are morphologically and functionally normal

We next determined whether platelets generated in the heart-lung system display classical morphology and function. Platelets display an almost uniquely characteristic sub-plasma membrane microtubular ring, running circumferentially in resting platelets[20,21]. Our generated platelets, immunolabelled for α-tubulin, display this characteristic ring structure (Fig. 4a), and the mean size of the cells is larger than controls ($3.6 ± 0.2$ μm vs $1.9 ± 0.05$ μm, Fig. 4b). However, it is also clear that there appear to be two subpopulations of generated platelets, based on their diameter ranges as shown in Fig. 4b: ~33% of generated platelets (diameter range: 1.7−2.4 μm) have sizes similar to control platelets (diameter range: 1.2−2.4 μm) and 67% of generated platelets (diameter ranges: 3.7−5.6 μm) are significantly larger than control platelets. We next visualized the ultrastructure of generated platelets by transmission electron microscopy (TEM, Fig. 4c), after depletion of host platelets using anti-GPIbα antibodies. Generated platelets displayed a discoid shape with classical characteristics including α-granules, dense granules, mitochondria, open canalicular system, and microtubule coils.

We then determined the functionality of platelets generated in the mouse heart-lung system by comparing against control mouse

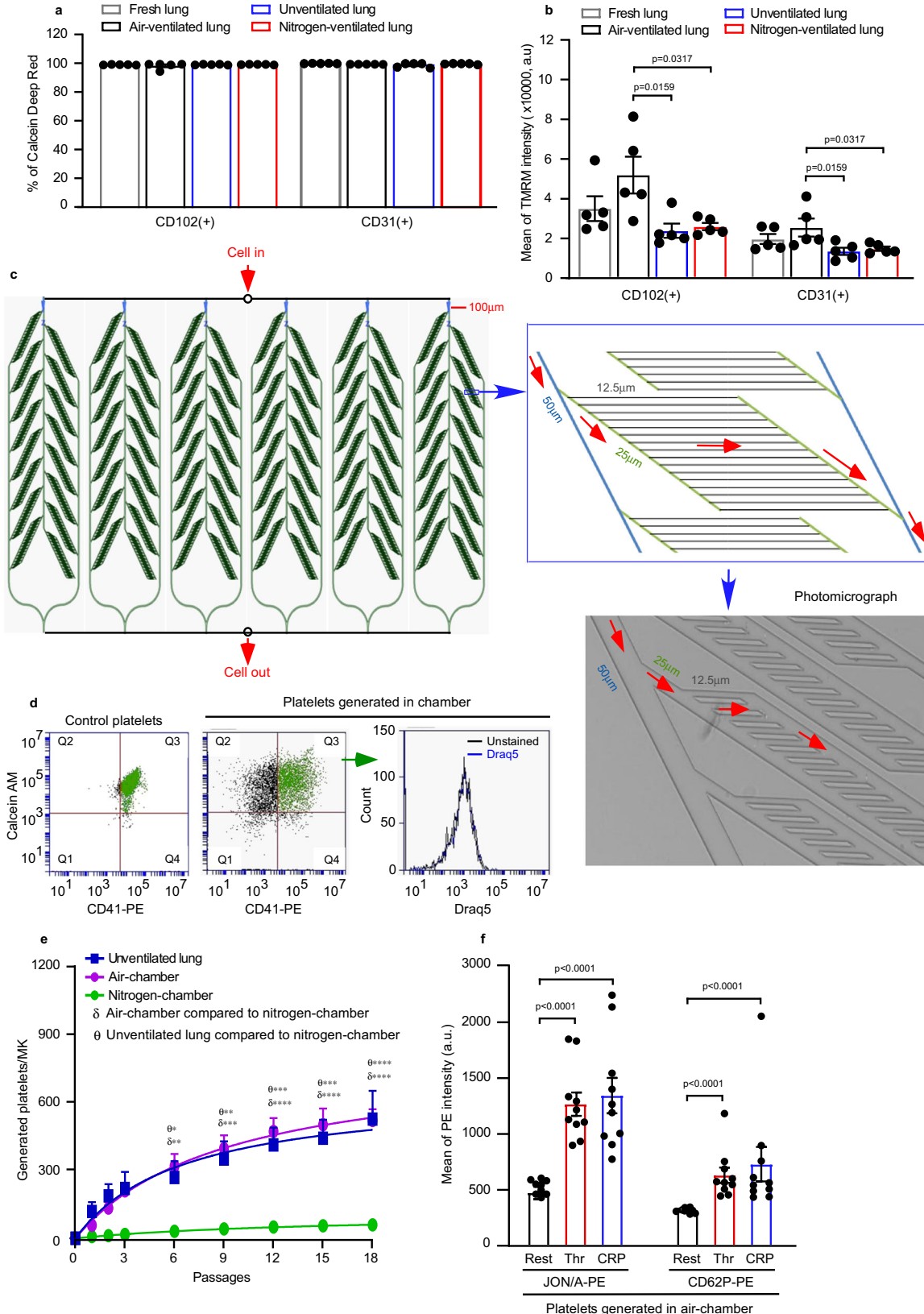

platelets. Both generated and control platelets showed equivalent responses to agonists (thrombin and CRP-XL) in terms of integrin αIIbβ3 activation and degranulation (P-selectin expression, Fig. 5a). Given that generated platelets appeared to segregate into two size subpopulations, we then compared the responses in these two subpopulations. The subpopulation with the larger size (diameter ranges:

3.7–5.6 μm) were more responsive, by comparison with the subpopulation with the smaller size, to thrombin in both integrin αIIbβ3 activation and P-selectin expression. It has been shown that larger platelets are more responsive[22,23], and our data are therefore consistent with this observation. We next compared the key glycoprotein expression on the surface of platelets generated in the mouse

**Fig. 3 | Role of pulmonary endothelial cell health and microvascular structure on platelet generation. a, b** Pulmonary endothelial cells (ECs) were isolated from perfused lungs under air- (black) or pure nitrogen-ventilation (red) or without ventilation (blue) for ~2 h. ECs from fresh lung tissue served as control (grey). ECs were stained with FITC-conjugated anti-CD31/PECAM-1 or anti-CD102/ICAM-2 antibodies. *N* = 5 independent experiments for each group. Two-tailed Mann−Whitney test. **a** Viability of pulmonary ECs were determined by Calcein Deep Red retention and displayed as mean ± SEM. **b** Mitochondrial membrane potential was determined by accumulation of Tetramethyl rhodamine methyl ester (TMRM) in active mitochondria and displayed as mean fluorescence intensity ± SEM. Air-ventilated lung vs unventilated lung: *p* = 0.0159 and 0.0159 for CD102 and CD31, respectively. Air-ventilated lung vs nitrogen-ventilated lung: *p* = 0.0317 and 0.0317 for CD102 and CD31, respectively. **c** Design of microfluidic chamber simulating a physiological pulmonary vascular system (details shown in Methods), including a photomicrograph of the smallest channels in the system and indication of the flow direction by arrows (red). Dimensions indicated on the figures are width of channels. **d, e** Mouse megakaryocytes (MKs) prelabelled with CD41-PE were repeatedly pumped through the microfluidic chamber. **d** The viability of generated platelets from the microfluidic chamber was determined by Calcein AM staining (Generated platelets: CD41+/Calcein AM+ in upper right quadrant, Q3 in green). All generated platelets identified in this way showed no DNA content (DRAQ5 -ve staining). **e** Quantification of generated platelets per MK in perfusates under air (purple circles) or pure nitrogen conditions (green circles), measured by FACS. For comparison, numbers of platelets generated in the unventilated lung-heart system (blue squares), from Fig. 2a, are shown. *N* = 4 (for air-chamber or nitrogen-chamber) and 5 (for unventilated lung-heart model) independent experiments. Data are mean ± SEM. Two-way ANOVA with Tukey's multiple comparisons test, *p < 0.05, **p < 0.01, ***p < 0.001 and ****p < 0.0001. **f** Mouse MKs prelabelled with CD41-FITC were repeatedly pumped through the microfluidic chamber under air 18 times. Generated platelets were washed and then integrin αIIbβ3 activation and P-selectin expression, induced by 2 U/mL thrombin (Thr) or 5 μg/mL CRP-XL, were measured by FACS. *N* = 10 independent experiments and data are mean ± SEM. Two-tailed Mann−Whitney test, *p* < 0.0001. Source data are provided in the Source Data file.

heart-lung system. The mean fluorescence intensity (MFI) of both CD61 and CD42b were comparable between generated and control platelets (Fig. 5b). However, whilst the proportion of cells expressing CD61 was also comparable, the proportion of cells expressing CD42b was lower in generated platelets compared to controls (Supplementary Fig. 1g). The MFI of three collagen receptors CD42d[24], CD49b and Glycoprotein VI (GPVI)[25] was higher in generated platelets than controls (Fig. 5b), while the proportion of cells expressing these receptors was lower (Supplementary Fig. 1g). It has been reported that surface expression of CD61, CD42b, CD49b and GPVI were higher in larger platelets, commensurate with their larger surface area[26]. The subpopulation of generated platelets with the larger size has higher surface expression of CD61, CD42b, CD42d, CD49b and GPVI by comparison with the subpopulation with the smaller size (Supplementary Fig. 1h).

Thrombus formation in vitro was also assessed, determining how generated platelets mixed into whole blood interact with a collagen-coated surface under flow. Generated platelets (stained with both DiOC6 and CellTracker™ Red CMTPX dye, blue) occupied all levels of the thrombus whilst control platelets (stained with CellTracker™ Red CMTPX dye alone, magenta) were mainly situated on top of the thrombus, suggesting generated platelets showed a higher responsiveness to collagen, or were primary reactors to it (Fig. 5c−e and Supplementary Movie 6). This may suggest that generated platelets are early interactors with collagen, displaying the greater adhesive functionality of younger platelets[22,23], possibly due to higher levels of collagen adhesive receptors (GPVI and CD49b[25], and CD42d[24]) (Fig. 5b).

**Megakaryocytes undergo enucleation and platelet release intravascularly**

We wanted to explore the details of the release of platelets from MKs upon repeated passage through the pulmonary vasculature. The cells in the perfusate were imaged after collected over a defined numbers of passages (0, 1, 2, 3, 6 and 9) (Supplementary Fig. 1a), and strikingly, upon repeated passages, MKs gradually move their nuclei to the periphery and subsequently enucleate, generating both naked nuclei and enucleated MKs. Although small numbers of enucleated round MKs were found, we saw the gradual accumulation of larger anuclear objects (>10 μm). Figures 1b, 6 and Supplementary Movies 7−10 show the steps involved in the process, with images shown in Fig. 6a, quantified in Figs. 1b and 6b. As shown in Fig. 1b, the percentage of intact MKs decreased from 53.7% after 1 passage (P1) to 6.7% after 9 passages (P9), while in Fig. 6b the percentage of large naked nuclei (>20 μm diameter) increased from 5.0% after P1 to 22.0% after P9. At the same time, large anuclear objects (>10 μm) also increased, as a proportion of total MKs and derivatives, from 16.3% after P1 to 45.5% after P9. These data suggested that the large polyploid nucleus moves from a central position to the periphery of the cell, in a process of

polarization. The nucleus is then extruded from the cell upon further passages through the lung vasculature, until by ~9 passages there are very few nucleated MKs left. After 12 passages, large naked nuclei (>20 μm) also became rare, being replaced by irregular small sub-nuclei. These sub-nuclei appeared connected to each other, probably by membranous structures or thin DNA bridges[27] that have been described previously between segregated chromosomes which were too fine for visualization by light microscopy (Fig. 6c and Supplementary Movie 11). Sub-nuclei were characterized by depth (Fig. 6d), aspect ratio (width: height, Fig. 6e), major axis (Fig. 6f) and minor axis (Fig. 6g) all of which decreased substantially with increasing passages (from P3 to P18, parameters: depth 8.9−5.5 μm, aspect ratio 1.8−1.1, major axis 14.1−6.5 μm, minor axis 8.1−5.7 μm). The extruded naked nuclei therefore undergo a process of division into multiple component sub-nuclei, which proceed to condense into compact sub-nuclear units with a greater circularity. The anuclate MK proceeds to fragment into platelets after multiple passages, to reach plateau numbers by 15−18 passages (Fig. 2a). This is the first time these enucleating behaviours have been observed for MKs when releasing platelets.

**Tropomyosin 4 is required for platelet generation in lung vasculature**

The release of platelets is understood to require cytoskeletal reorganization involving the actin cytoskeleton. Tropomyosins form co-polymers with actin filaments and regulate filament function in an isoform-specific manner[28]. TPM4 has been shown to have a role in platelet formation. In the *Tpm4*−/− mouse in vivo, platelet counts drop by ~35% compared to wild-type and with a slightly larger mean platelet volume[29]. Strikingly however, in our ex vivo lung system, *Tpm4*−/− MKs generate no platelets in the perfusates (Fig. 7a, b). We therefore wanted to explore the steps in platelet generation requiring TPM4. Compared to wild-type MKs, during the first 3 passages, fewer *Tpm4*−/− MKs underwent transformation to large anuclear objects (Fig. 7c). However, this represented only a delay in these events, since by 6−9 passages the numbers of large anuclear objects were comparable with wild-type (Fig. 6b). Two-photon microscopy of fixed lung sections, after 18 passages, showed abundant anuclear fluorescent objects, sized ~10 μm (Fig. 7d and Supplementary Movie 12). These data therefore suggest a small and non-essential role for TPM4 in regulating enucleation, but a critical role in regulating the final steps of platelet formation and their release into the circulation. This unexpected and striking observation of the critical role for TPM4 in platelet generation in our heart-lung model, together with the observation that *Tpm4*−/− mice still have 65% of normal platelet numbers, caused us to investigate whether *Tpm4*−/− MKs behave similarly to WT in the bone marrow. We observed the 4 different MK morphologies described by Junt et al.[30] in the progression of MKs from marrow space to sinusoid equally in WT or *Tpm4*−/− mice

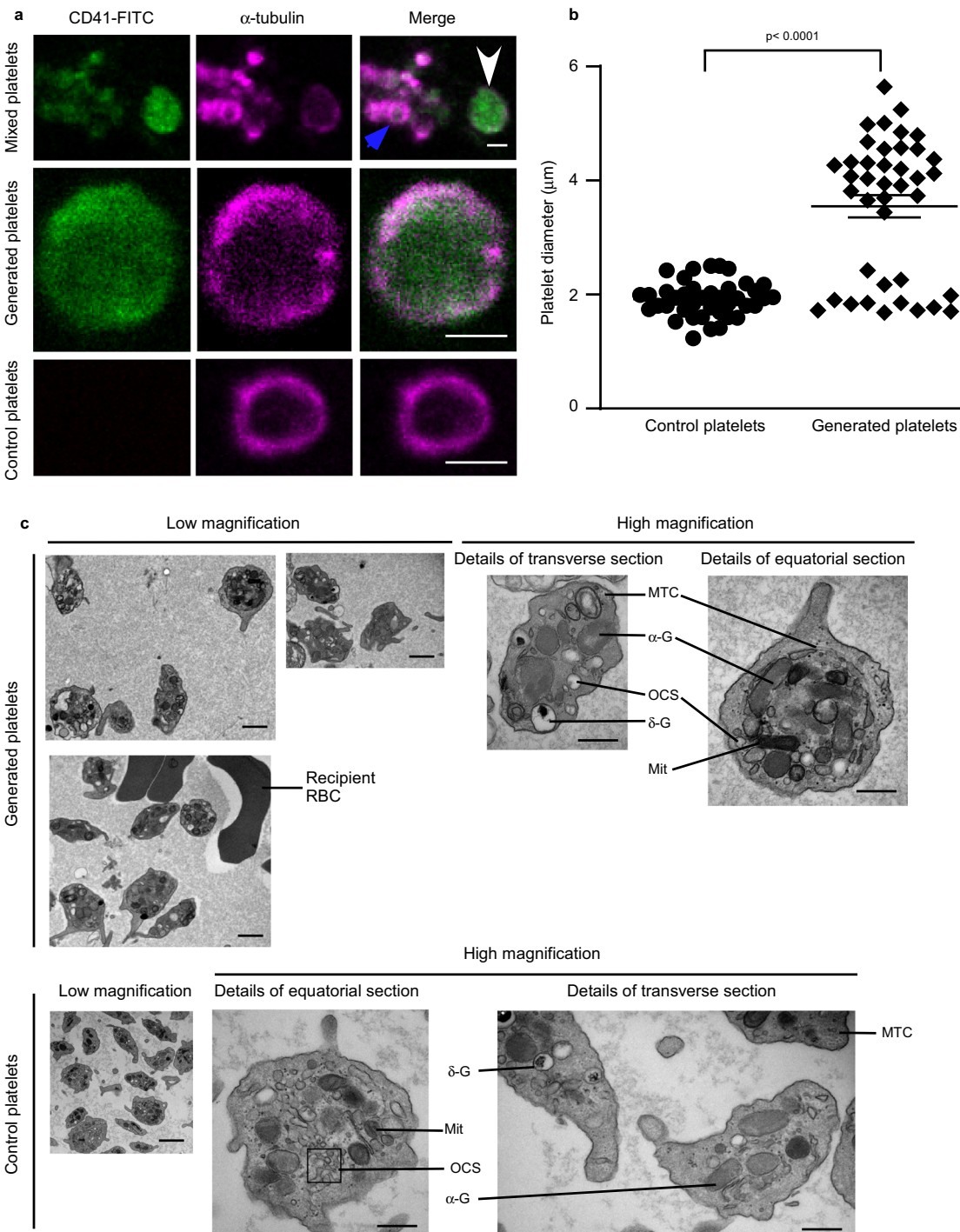

**Fig. 4 | Generated platelets demonstrate typical physical features comparable to control platelets. a**, **b** Mouse megakaryocytes (MKs), prelabelled with CD41-FITC (green), were passaged repeatedly through the pulmonary vasculature ex vivo. Lungs were ventilated with air throughout. **a** Perfusates from ex vivo heart-lung preparation, containing both generated platelets (white arrow) and host platelets (blue arrow), were stained for α-tubulin (magenta), and confocal images shown as a mixed population in the top panels. More detailed images of a-tubulin rings are shown in the magnified images in the middle panel (generated platelets) and bottom panel (control platelets). Images are representative of $n = 3$ independent experiments. Scale bars: 2 μm. **b** The diameter of platelets (40 platelets from $n = 3$ independent experiments) from **a** was measured using Fiji ImageJ-Win64, with diameters of generated platelets: 3.6 ± 0.2 μm vs control platelets: 1.9 ± 0.05 μm. In contrast to control platelets, there were two subpopulations of generated platelets based on their diameter ranges: ~33% of generated platelets (diameter range:

1.7–2.4 μm) have sizes similar to control platelets (diameter range: 1.2–2.4 μm) and 67% of generated platelets (diameter ranges: 3.7–5.6 μm) are significantly larger than control platelets. Data are mean ± SEM. Two-tailed Mann–Whitney test, $P < 0.0001$. **c** Ultrastructures of generated platelets and control platelets visualized by transmission electron microscopy. Host platelets were depleted by intraperitoneal administration of anti-GPIbα antibody R300 prior to perfusing MKs through the heart-lung preparation under air-ventilation. Subcellular structures are shown and annotated as abbreviations, in the high magnification images: α-G, α-granules; σ-G, σ-granules or dense bodies; Mit, mitochondria; OCS, open canalicular system; MTC, microtubule coils; RBC, red blood cells. Scale bars: 2 μm in the images with low magnification, 500 nm in the images with high magnification. Images shown are representative of $n = 5$ independent experiments. Source data are provided in the Source Data file.

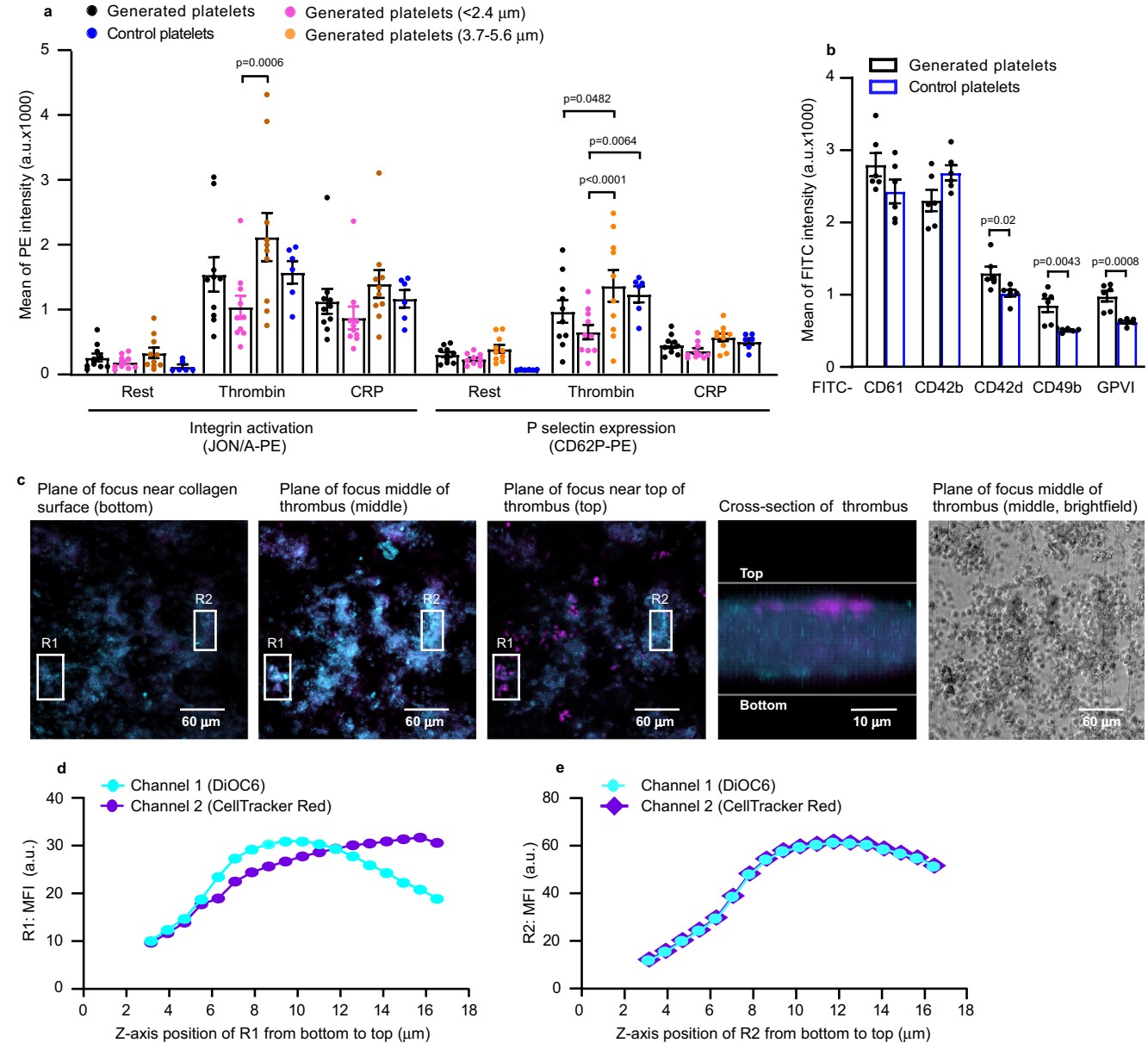

**Fig. 5 | Generated platelets are functionality comparable to control platelets.** Mouse megakaryocytes (MKs), labelled with CD41-FITC or CD41-PE antibody or DiOC6 dye, were passaged through pulmonary vasculature ex vivo 18 times. Lungs were ventilated with air throughout. **a** Mean fluorescence intensity (MFI), measured by FACS, of integrin αIIbβ3 activation and P-selectin expression induced by 2 U/mL thrombin or 5 μg/mL CRP-XL in washed generated platelets (CD41-FITC) compared to washed control platelets. Data from generated platelets were pooled as total generated platelets (black dots), and segregated by platelet size (diameter <2.4 μm as pink dots, diameter 3.7–5.6 μm as light brown dots), compared to control platelets (blue dots). Generated platelets sizes were estimated using Particle Size Standard Kit (Cat. PPS-6). $N = 10$ (generated platelets) and 6 (control platelets) independent experiments and data are mean ± SEM. Two-way ANOVA with Tukey's multiple comparisons test, $p$-values indicated in the figure. **b** Surface glycoproteins were measured by FACS. Generated platelets were defined by staining with anti-CD41-PE antibody. Surface glycoproteins were stained with different FITC-conjugated antibodies as indicated. $N = 6$ independent experiments and data are

mean ± SEM of FITC intensities. Two-tailed unpaired $t$-test, $p = 0.02$ for CD 42d, $p = 0.0043$ for CD49b and $p = 0.0008$ for GPVI. **c** Images of representative platelet-rich thrombus. Generated platelets (showing as blue in colour, stained with both DiOC6 (cyan) and CellTracker™ Red CMTPX dye (magenta)) occupied all levels of the thrombus while host platelets (stained with CellTracker™ Red CMTPX dye alone, magenta) were mainly situated on top of thrombus. Images are representative of $n = 5$ independent experiments. Scale bars as indicated. **d**, **e** MFI profiles of R1 and R2 (Region 1 and Region 2 from Fig. 5c) along the z-axis. **d** MFI profile of R1 along the z-axis. In R1, generated platelets occupied the lower part of the thrombus up to ~12 μm (both cyan and magenta signals increased simultaneously), whilst beyond this point host platelets were predominant (as magenta signals were stronger than green beyond 12 μm). **e** MFI profile of R2 along the z-axis. In R2, this part of thrombus was composed only of generated platelets as both cyan and magenta signals changed simultaneously along z-axis. Source data are provided in the Source Data file.

(Fig. 8 and Supplementary Movies 13–14): (i) the majority of *Tpm4*$^{-/-}$ MKs were seen as isolated cells within the bone marrow space, in close contact with sinusoidal walls; (ii) some MKs within the marrow space produced extensions into sinusoids; (iii) some MKs were clearly visible wholly within the sinusoid vessels themselves and had cellular

extensions; (iv) some appeared as large fragments in the sinusoid, releasing heterogeneous structures in the direction of blood flow. These data may suggest that in the lung vasculature, TPM4 regulates transformation of ~10 μm anuclear fragments into platelets, and therefore no platelets could be produced in lung-heart system. In

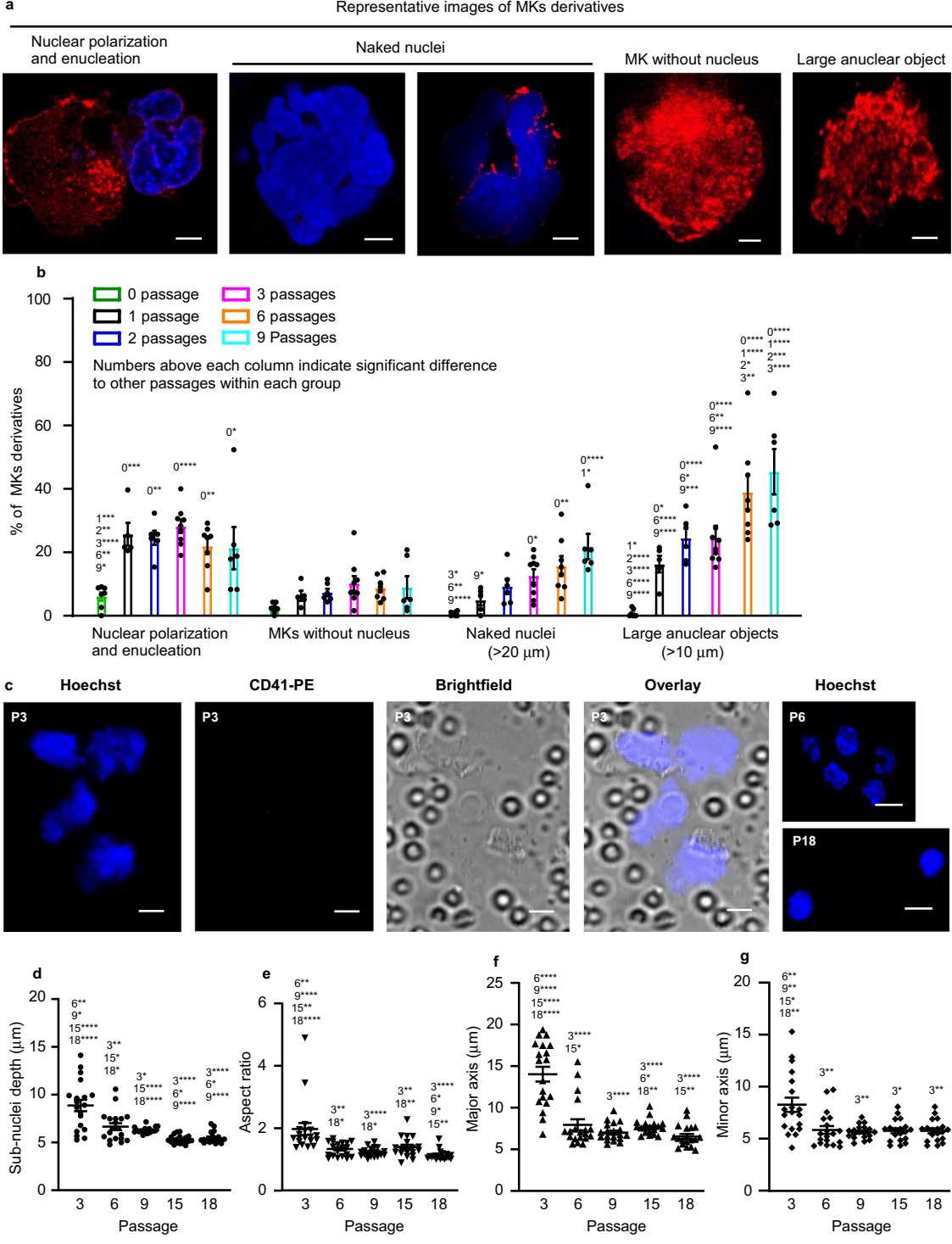

addition, TPM4 may play some role in regulating the final size of platelets generated in organs other than the vasculature of the lung, as the platelet volume in $Tpm4^{-/-}$ mouse is slightly larger than wild-type mice.

## Discussion

In this study, we established an ex vivo pulmonary vascular model, which we show generates platelets from cultured MKs outside the body with an output similar to estimate in vivo MK capabilities. These generated platelets display classical morphology such as α-tubulin ring, α-granules, dense granules, mitochondria, and microtubule coils and show comparable functionality to native platelets. The ex vivo model therefore provides a useful tool to efficiently generate platelets

and explore the mechanisms of platelet generation, and also demonstrates the capability of the lung as a site of platelet generation.

The study presented makes several advances in our understanding of the mechanisms of platelet generation in lung vasculature. First, our data show that four factors may affect platelet generation from MKs in lung vasculature: oxygenation, ventilation, healthy pulmonary endothelium and the microvascular structure. These factors suggest the lung is a unique site for platelet biogenesis. This may partially explain the range of platelet counts reported in patients with lung disease[31,32] or in people living at altitude[33,34]. The wide range of platelet counts may be because the process of platelet generation in the lung is complex, with inputs from the four factors we describe and

**Fig. 6 | Megakaryocytes show nuclear marginalization and enucleation prior to fragmentation.** Megakaryocytes (MKs) from C57BL/6 mice, labelled with CD41-PE antibody (red) and Hoechst 33342 (blue), were passaged repeatedly through the pulmonary vasculature of a C57BL/6 mouse ex vivo. Lungs were ventilated with air throughout. **a** Representative images of MKs derivatives during platelet generation: nuclear polarization and enucleation, where the nucleus is marginalized, of irregular shape or in the process of ejection from the cell; naked nuclei, where the ejected nucleus is >20 μm in diameter and free from the parent cell and/or partially encased in thin/patchy plasma membrane; MKs without nucleus, where the MKs have an approximately circular shape but without nuclei; and large anuclear objects, where ghost cells are of irregular shape and >10 μm in their longer axis. Scale bar: 5 μm. **b** Cells were imaged from samples of perfusates after passage numbers 1, 2, 3, 6 and 9. Five subgroups of MKs and their derivatives, as described above, were quantified as a percentage of total number of cells. Quantification was from at least 250 fields of view, counting at least 230 cells in total for each of the

subgroups. $N = 7, 5, 6, 9, 8$ and 6 independent experiments for passage numbers 0, 1, 2, 3, 6 & 9. Data are mean ± SEM. Two-way ANOVA with Tukey's multiple comparisons test, *$p < 0.05$, **$p < 0.01$, ***$p < 0.001$ and ****$p < 0.0001$. **c–g** Nuclear lobes of MKs fragment into small condensed sub-nuclei. **c** Representative images, from $n = 3$ independent experiments, of naked nuclei generated after multiple passages (as indicated) of mouse MKs through pulmonary vasculature. Scale bars: 5 μm. **d–g** The depth (**d**), aspect ratio (**e**), major axis (**f**) and minor axis (**g**) of sub-nuclei decreased substantially with increasing passages. These parameters were measured from after 3 passages to after 18 passages: depth 8.9–5.5 μm, aspect ratio 1.8–1.1, major axis 14.1–6.5 μm, minor axis 8.1–5.7 μm. Numbers above each column indicate significant difference to other passages. 19 sub-nuclei from $n = 3$ independent experiments were collected. Data are mean ± SEM. Two-tailed Mann–Whitney test, *$p < 0.05$, **$p < 0.01$, ***$p < 0.001$ and ****$p < 0.0001$. Source data are provided in the Source Data file.

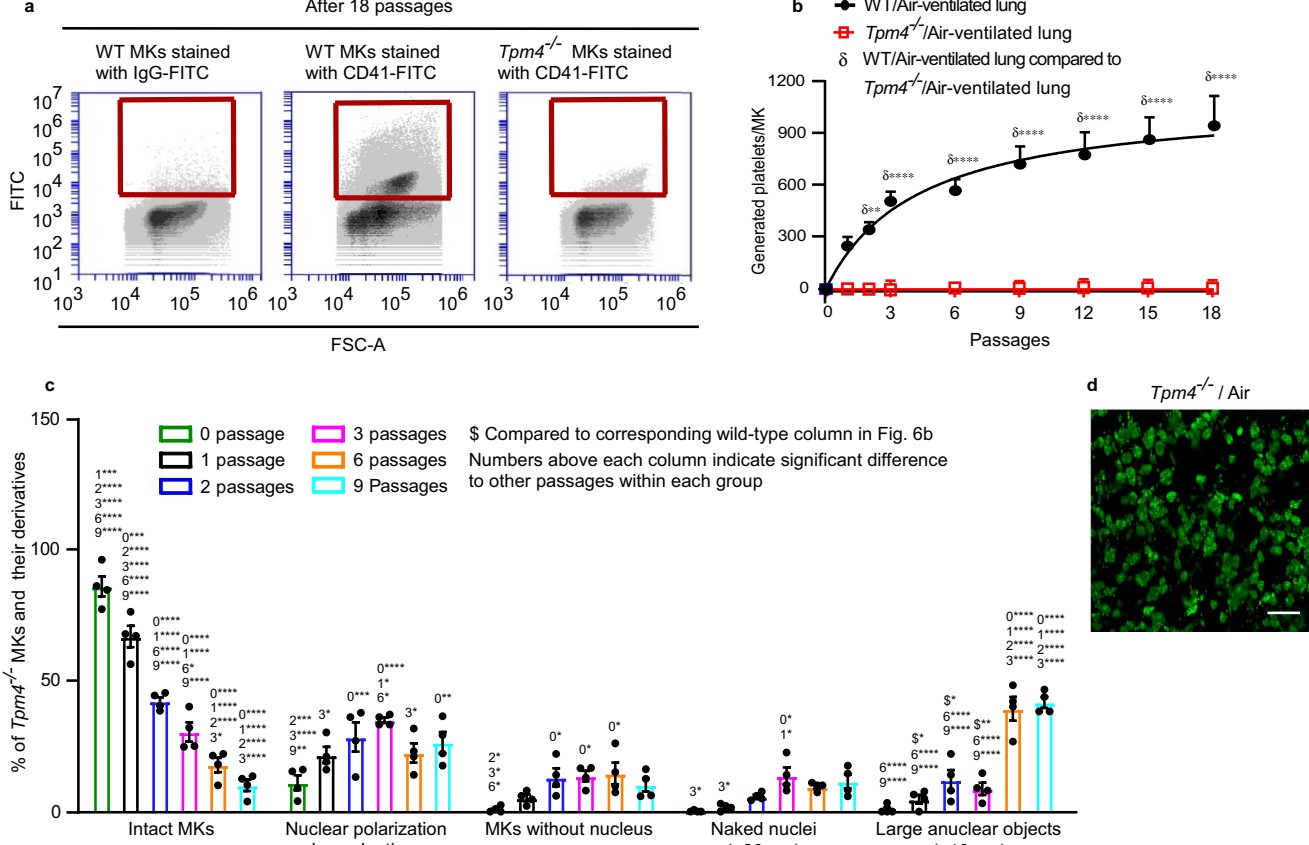

**Fig. 7 | Tropomyosin 4 is required for the final steps in platelet generation in the pulmonary vasculature.** Megakaryocytes (MKs) from *Tropomyosin4*$^{-/-}$ (*Tpm4*$^{-/-}$) mice labelled with FITC-conjugated anti-CD41 antibody, were passaged repeatedly through the pulmonary vasculature of a C57BL/6 mouse ex vivo. For controls, wild-type (WT) MKs stained either with FITC-conjugated anti-CD41 or isotype antibodies, were passaged repeatedly through the pulmonary vasculature of a C57BL/6 mouse ex vivo. Lungs were ventilated with air throughout. **a** Representative FACS dot plot images are shown for generated platelets (CD41-FITC positive events are within the red square). **b** Numbers of generated platelets per *Tpm4*$^{-/-}$ MKs, or control WT MKs, in perfusates after different passage numbers through WT lung, were quantified by FACS. *Tpm4*$^{-/-}$ platelets were consistently undetectable after up to 18 passages, in the perfusate. Data shown are platelets generated per MK from either *Tpm4*$^{-/-}$ MKs or control WT MKs and displayed as mean ± SEM. $N = 5$ independent experiments for each group. Two-way ANOVA with Šídák's multiple comparisons test, *$p < 0.05$, **$p < 0.01$, ***$p < 0.001$ and ****$p < 0.0001$. **c** Cells were imaged from samples of perfusates after passage numbers 0, 1, 2, 3, 6 and 9 through

murine lung vasculature ex vivo. Cells were morphologically classified as 5 subgroups: intact MKs (as per Fig. 1b) and MK derivatives (shown in Fig. 6a) and quantified as a percentage of total number of cells. Quantification was from at least 150 fields of view, counting at least 170 cells in total for each of the subgroups. $N = 4$ independent experiments and data are mean ± SEM. Numbers above each column indicate significant difference to other passages within each group. Two-way ANOVA with Tukey's multiple comparisons test, *$p < 0.05$, **$p < 0.01$, ***$p < 0.001$ and ****$p < 0.0001$. Dollar sign ($) above columns represents significant difference to corresponding wild-type column in Fig. 6b. Two-tailed unpaired t test (passages 1, 2, 6 and 9) or Mann–Whitney test (passage 3), *$p < 0.05$, **$p < 0.01$. **d** Abundant fluorescent objects, ~10 μm diameter, were visible in sections of mouse lung after 18 passages of stained *Tpm4*$^{-/-}$ MKs, as shown in extended focus stacks of 20 continuous two-photon planes of lung. Images shown are representative of $n = 3$ independent experiments. Scale bar: 20 μm. Source data are provided in the Source Data file.

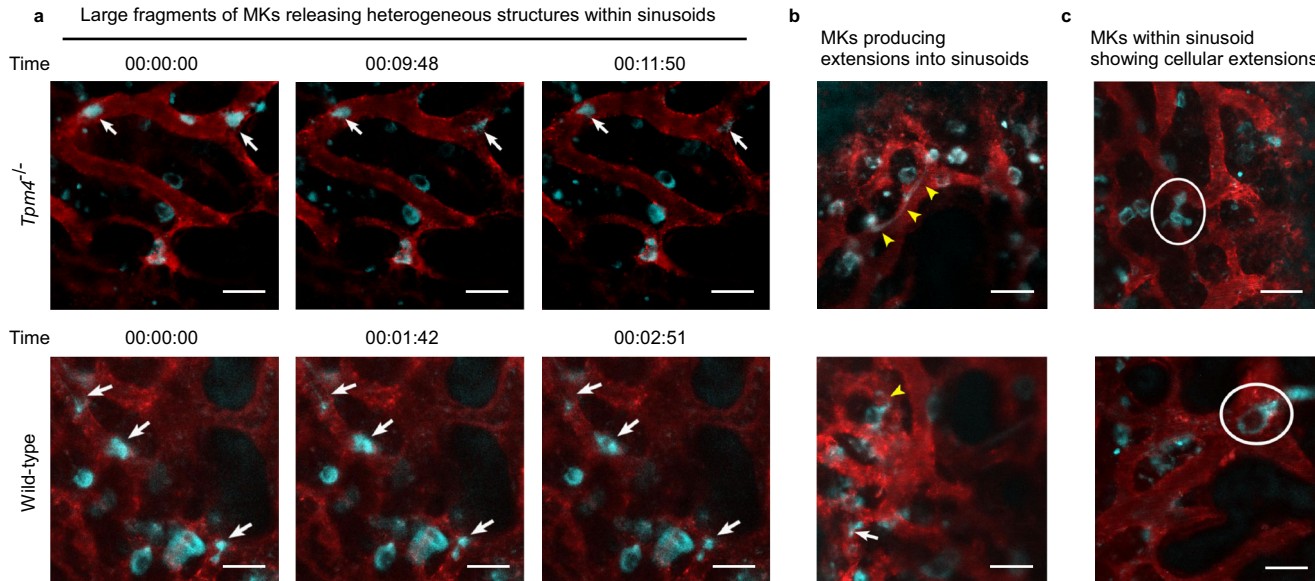

**Fig. 8 | Intravital two-photon microscopy of bone marrow megakaryocytes in live mouse calvarium.** Bone marrow vasculature was visualized by intravenous injection of anti-CD105-AlexaFluor 546 antibody and AlexaFluor 546-labelled BSA (red). Megakaryocytes (MKs) and their derivatives were stained intravenously with anti-GPIX-AlexaFluor 488 antibody (cyan). Both wild-type (WT) and *Tropomyosin 4[-/-]* (*Tpm4[-/-]*) MKs were generally seen in close contact with the bone marrow sinusoidal walls. Images were taken from *n* = 6 biologically independent mice for each group (WT and *Tpm4[-/-]*). Scale bars, 50 μm. **a** Representative images of large fragments of WT and *Tpm4[-/-]* MKs (white arrows) within sinusoidal vessels releasing heterogeneous structures in the direction of blood flow, at time points indicated. **b** Representative images of WT and *Tpm4[-/-]* MKs, with cell bodies within the marrow space, producing extensions into sinusoids (yellow arrowheads). **c** Representative images of WT and *Tpm4[-/-]* MKs within the sinusoid, showing cellular extensions (white circles).

possible redundancy or partial redundancy between these four factors. For example, there may be compensation for low oxygenation by an increase in respiratory rate.

Second, in this study, we found that MKs have a substantial ability to reversibly deform to passage through the lung microvasculature, despite the mismatch in size of giant MKs (50–100 μm)[5] and the narrow internal diameter of pulmonary capillaries (mean 5–8 μm[19]) which further decrease upon lung inflation[35]. We and others have shown that intact mature MKs can egress from the bone and enter the circulation[6,36,37] whereupon they find their way to the first microvascular bed in the lung[38,39]. It has been observed that they will lodge in the pulmonary vasculature where they release platelets, and consistent with this substantial numbers of MKs with reduced cytoplasmic content or no visible cytoplasm (denuded MKs) have been found in aortic circulation[37,39]. However, whether intact mature MKs can squeeze through the pulmonary circulation and enter the left side of the heart and then the arterial circulation remains unclear. Our data show ~50% of infused intact mature MKs (with circular shape and central nuclei) are able to pass through the lung microvasculature after the first passage through the lung in the ex vivo model (Fig. 1b). We also demonstrated that intact mouse MKs could pass through the pulmonary vasculature in vivo and appear in the blood of the left common carotid artery of an anaesthetized recipient C57BL/6 after being infused into the right external jugular vein (Fig. 1c). Interestingly, mature MKs with abundant, finely granular cytoplasm and compact lobulated nucleus have been observed in peripheral blood smears[40–42], consistent with our findings. These large cells were usually found at the feathered edge of the peripheral blood smear. Although it is possible that some intact MKs pass through the lung circulation via physiological shunts, these only account for ~2% of blood flow through the lung, and therefore the majority of MKs are likely to passage through the capillary bed of the lung.

The ability of MKs to pass through the lung microvasculature is oxygen-dependent in our ex vivo system, as longer term exposure of the lung to pure nitrogen, to completely de-oxygenate the lung over 2 h, effectively caused MKs to be retained in the lung vasculature. Lack of oxygen may affect the platelet generation process through effects on the pulmonary vascular endothelium, or directly on MKs, or some interaction between both cell types. Oxygen-dependent MK motility might partially explain why pulmonary MK levels observed at autopsy are increased in COVID-19 patients who had died with acute lung damage[43].

Third, we found that giant nuclei extrude from MKs over the process of multiple passages, as part of the process leading to platelet generation, and then divide into multiple component sub-nuclei, and further condense into compact sub-nuclear units. This observation is consistent with the phenomenon that denuded MKs have been found in both the bone marrow and peripheral circulation[37,44]. The extruded naked giant nuclei are rapidly removed by the mononuclear phagocyte system[37] in healthy individuals, but become apparent in people with impaired immunity such as in patients with human immunodeficiency virus (HIV)[45] and in the lungs of severe Covid-19 patients[44]. Few cells are known to enucleate, but importantly these include erythrocytes. This may be important because MKs and erythrocytes are developmentally closely related, sharing a common precursor, the MK/erythroid progenitor (MEP)[46,47], and may therefore indicate a common mechanism.

The prevalent model for thrombogenesis, extension of proplatelets and their detachment under flow, proposes that MKs extend long (>100 μm) branched processes that appear "beaded" by virtue of intermediate swellings[48], and then undergo reorganization into platelets. However, in our intravascular model the data show that after extruding their nuclei, MKs generate smaller and smaller fragments, eventually forming platelets. Mature MKs first fragmenting into large anuclear structures in our heart-lung model is consistent with Junt's study in bone marrow in vivo which showed MKs shedding large cytoplasmic fragments at the vascular niche in the bone marrow and almost all MK fragments releasing into the vasculature were 10–100 times as large as circulating platelets[30]. Some of these elements may well be proplatelets, which form an intermediate structure on the path from large anuclear MKs finally to generate platelets.

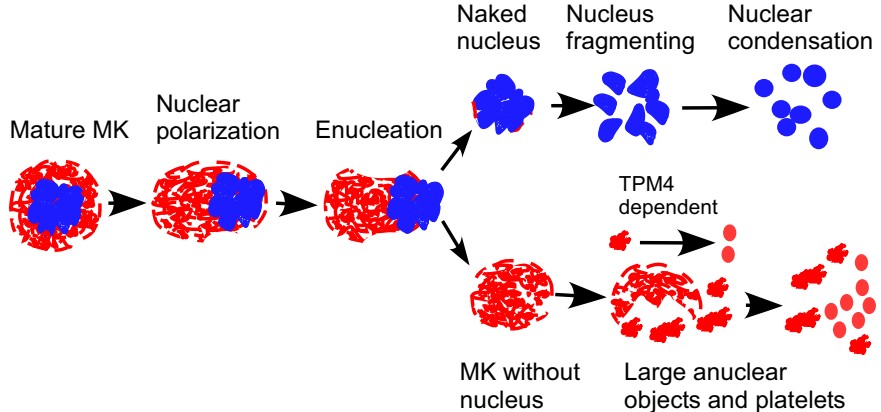

**Fig. 9 | Schematic diagram of the steps in platelet generation from mature megakaryocytes.** Diagram showing platelet generation pathway from intact mature megakaryocytes (MKs) to final platelet formation, by repeated passage of MKs through the pulmonary vasculature. The process involved nuclear polarization, enucleation, gradual cytoplasmic fragmentation into platelets (with the final steps dependent on Tropomyosin 4 (TPM4)) and nuclear fragmentation and condensation.

Fourth, multiple passage of MKs through the pulmonary vasculature in the ex vivo lung model is crucial to induce a reproducible sequence of events (Fig. 9). These include enucleation, nuclear fragmentation and condensation, prior to efficient platelet generation. This might suggest an essential and dynamic conversation between the pulmonary microvasculature and MKs, a process effectively of 'education'. This education is required to induce reversible deformation of MKs, stimulate their motility through the microvasculature and drive platelets release. However, since some MKs pass through the pulmonary vasculature, it is possible that some platelet generation may take place in other organs of the body, such as spleen[49]. This is supported by the observation that we can generate substantial numbers of platelets by passage of MKs through the microfluidic chamber that simulates the microvasculature. This in vitro system lacks endothelium and physical ventilation but retains an architecture that approximates the microvasculature. Production of platelets in the microfluidic chamber is about one sixth of that produced in the ex vivo lung model (492.3 platelets per MK by microfluidic chamber vs 2997.0 per MK generated in pulmonary vascular model) (Figs. 2c and 3e). This suggests that there is synergy between the features of the lung vasculature that makes this organ likely to be an important site of vascular production of platelets, but also suggests that platelet production can still take place, less efficiently, if not all features are present.

The lung is an important site of platelet generation, with reports of production ranging from 7–17% of total body platelets[17] to 50%[13]. Mice lacking *TPM4* in the MK lineage display macrothrombocytopenia, with ~35% decrease in platelet number[29]. In our ex vivo lung system, despite undetectable platelets in the perfusate (Fig. 7a, b), abundant anuclear fluorescent objects, sized ~10 μm, could be seen in the lung vasculature (Fig. 7d and Supplementary Movie 12). This suggests that TPM4 is required for the final steps in platelet generation in the lung vasculature. Importantly, our data from intravital observation of *Tpm4⁻/⁻* bone marrow suggests that MK protrusion/extrusion process, which may be proplatelet formation in vivo, is similar to normal (Fig. 8 and Supplementary Movies 13–14) apart from the 1.2-fold increase in platelet volume. The data suggest that TPM4 plays an essential role in the transformation of large (~10 μm diameter) anuclear MK fragments to platelets in the lung vasculature. Outside of the lung however TPM4 may only play a role to regulate final platelet size, since *Tpm4⁻/⁻* platelets are ~1.2-fold larger than wild-type. These data also imply there may be different mechanisms in the final steps of platelet release in different organs of the body. It is also possible that the lower platelet count seen in *Tpm4⁻/⁻* mice is a product of defective platelet formation outside of the bone marrow, in the lung vasculature.

Finally, the work introduces a system for generating platelets in vitro, by multiple passage of MKs through a microfluidic chamber. Current systems developed to generate platelets in vitro include 2D and 3D culture systems, 3D bioreactors and big tank bioreactors, including the use of turbulence to generate platelets from MKs[50–52]. Thon et al. developed a microfluidic system that applies shear to emergent proplatelets from MKs[53]. Our system differs in that cells are required to pass through small channels, equivalent to capillaries, with internal dimensions 12.5 × 10 μm. The approach is therefore mechanistically different to other systems and is capable of generating large numbers of platelets, ~500 platelets per MK, although it is likely that further improvements will be forthcoming, including refinements using endothelial cells in the system.

In summary, this work identifies a highly efficient mechanism for platelet generation outside of the body, by repeated passage of MKs through lung vasculature under air ventilation, involving enucleation and final TPM4-dependent steps to generate platelets. The findings will inform new approaches, such as the microfluidic system reported here, to large scale generation of human platelets.

## Methods
### Animal experiments and ethics statement
C57BL/6 mice were purchased from Harlan UK. The study protocol of C57BL/6 mouse care and experiments was approved by the University of Bristol local research ethics committee (AWERB) and licensed under UK Home Office project license PPL 30/3445 and PP5643338. *Tpm4⁻/⁻* mice were generated as we have previously described[29]. The study protocol of *Tpm4⁻/⁻* mice was approved by the district government of lower Franconia, Germany (Bezirksregierung Unterfranken). Age and sex matched mice were used for each experiment. Both female and male mice were used at an age of 8–12 weeks (no selection for sex of mice). The mice were maintained in cages with ad libitum access to water and standard food, which were located in a 12 h light/dark cycle animal room with room temperature ~22 °C, and humidity of 40–60%.

### Mouse ex vivo heart-lung preparation
Mice were sacrificed by a Schedule 1 process, by exposure to rising $CO_2$. A tracheostomy was then performed, and lungs were ventilated with room air or pure nitrogen via a rodent ventilator (Minivent type 845, Harvard Apparatus, USA) or left unventilated. To avoid end-expiratory lung collapse, as indicated in Fig. 1a, the connector for expiration air was connected to a Gottlieb valve (tube immersed in water; immersion ~2 cm depth to induce a positive end-expiratory pressure). Respiratory rate was maintained at around 150–200

breaths/min and tidal volume was 10 mL/kg (~250 μL). The inferior vena cava was then exposed by laparoscopy, and the chest opened by median sternotomy and fat tissue carefully removed. A 21-gauge catheter was passed into the right ventricle. About 100 mL warm (36–38 °C) perfusion buffer (Krebs-Hepes buffer (KHB): 140 mM NaCl, 3.6 mM KCl, 0.5 mM NaH$_2$PO$_4$, 0.2 mM MgSO$_4$, 1.5 mM CaCl$_2$, 10 mM Hepes (pH 7.4), 2 mM NaHCO$_3$) containing 10 U/mL heparin was perfused at 3–4 mL/min into the lung vasculature. Once the colour of mouse lung turned pale, suggesting most of blood in the lung circulation was flushed, the catheter was disconnected.

The left and right superior venae cavae, ascending aorta, inferior vena cava and descending aorta were ligated using 6–0 silk, and a small incision was made in the left ventricle for collection of the perfusate. A suspension of mouse MKs prelabelled with a fluorescently tagged anti-CD41 antibody (containing 10 U/mL heparin, 1 mM ethylenediamine-tetraacetic acid (EDTA), 1 U/mL Apyrase and 100 ng/mL prostaglandin E1 (PGE1)) was placed into a perfusion pump and infused into the right ventricle, whilst the lungs were either ventilated with air or pure nitrogen or without ventilation, and the flow rate was maintained at 0.35 mL/min by a SyringeOne Programmable Syringe Pump (Product SKU: NE-1000-ES). This flow rate achieves a pressure (2.1–3.5 mmHg, measured by pressure transducer as shown in Fig. 1a) substantially lower than physiological levels of mouse pulmonary artery, which has been estimated to be ~25 mmHg[54].

The perfusate was collected and re-pumped into the right ventricle after samples were collected for imaging and fluorescence-activated cell sorting (FACS) analysis. The perfusate was recirculated through the pulmonary vasculature in this manner for a total of 18 passages. After the final passage, 1 mL perfusion buffer was pumped into the system to remove some of the remaining cells in the lung vasculature, and all perfusate collected for evaluation of platelet function. The mouse lung was immediately removed and the volume of the lung was measured by fluid displacement (Supplementary Fig. 1f). The lung was then fixed with 4% PFA/PBS at 4 °C overnight and kept from the light. Experimental flowchart for generating platelets from repeated infusion of mouse heart-lung preparation is shown in Supplementary Fig. 1a.

## Microfluidic chamber design, fabrication, and experimental protocol

A set of channels simulating a physiological vascular system[19] was constructed using standard PDMS approach. The design shows a branching structure, such that as branching progresses, the channel diameter decreases by half. All channels are rectangular in cross-section and 10 μm deep, with the largest channels being 100 μm across, decreasing to the smallest channels which are 12.5 μm across. From each larger channel, 16 smaller channels branch off, allowing for maintenance of flow resistance due to the $r^4$ power relationship between resistance and channel diameter. Fluid then flows from larger diameter channels to smaller diameter channels, and in reverse on the way out of the system. Cells are passed through the system, repeatedly. The system is scaled up, through multiplexing in parallel, to allow greater cell volumes to be used, as shown in Fig. 3c.

The mask and SU-8 master mold were fabricated by NuNano (Bristol, UK). The microfluidic channels were fabricated by soft lithography. The mixture of PDMS in a 10:1 ratio was poured over the SU-8 master mold after being degassed in a vacuum desiccator. The PDMS mixture was cured at 80 °C for 2 h and incubated in the oven overnight. The PDMS mold was removed from the SU-8 master. Input and output holes were punched using a 0.5 mm OD biopsy puncher (Elveflow, Paris, France). Finally, the PDMS microchannels were irreversibly bonded to glass slides using oxygen plasma for 3 min in a plasma device (Diener Plasma Systems, Ebhausen, Germany).

A suspension of mouse MKs prelabelled with CD41-PE or CD41-FITC antibodies (containing 1 mM EDTA, 1 U/mL Apyrase and 100 ng/ mL PGE1) was placed into a perfusion pump and infused into the microfluidic chamber at 0.30 mL/min flow rate. Suspensions were collected from the output hole and re-pumped into the microfluidic chamber, after 25 μL samples were taken for FACS analysis. Suspensions were then recirculated through the microfluidic chamber in this manner for a total of 18 passages.

## Culture and differentiation of murine megakaryocytes

Briefly, bone marrow from C57BL/6 or *Tpm4*$^{-/-}$ mice was isolated and dispersed prior to centrifugation at 200 × $g$ for 10 min. Bone marrow was re-suspended in IMDM- Glutamax containing 1% penicillin/ streptomycin and 2% serum replacement 1. Cells were cultured for 3 days in the presence of 20 ng/mL recombinant murine stem cell factor (rSCF) and a further 13 days in the presence of 10 ng/mL recombinant murine thrombopoietin (rTPO) at 37 °C and 5% CO$_2$. From day 8 to day 16, cells were transmitted to fresh cell culture dishes every day to reduce MKs to contact with fibroblast cells. On day 16, MKs were harvested and enriched on a 1.5%/3% bovine serum albumin (BSA) gradient for 1 h in cell incubator, following stained with either IgG-PE/ or FITC, or CD41-PE or -FITC for 3 h and further with DNA dye Hoechst 33342 for 20 min. Then MKs were washed and resuspended in 2 mL medium prior to use.

## In vivo passage of intact megakaryocytes through lung vasulature

C57BL/6 mice were anaesthetized by intraperitoneal injection of a 100 mg/kg ketamine/10 mg/mL xylazine mix. Mouse MKs were stained with CellTracker™ Red CMTPX dye and Hoechst 33342 prior to injection into the right external jugular vein of an anaesthetized recipient mouse. Blood was collected from the left common carotid artery. Cells were imaged by confocal fluorescence microscopy after lysis and removal of red blood cells.

## Flow cytometry

Platelets derived from IgG-PE or IgG-FITC stained MKs were set up as negative controls. Washed mouse platelets were used to set gates for generated platelets (P1 gate) at the FSC/SSC density plot by size and granularity (shown in Fig. 1d). 25 μL of CD41-PE or CD41-FITC stained MK suspension, or perfusates from lung or suspensions from microfluidic chambers after 1, 2, 3, 6, 9, 12, 15 or 18 passages, were measured by FACS. DNA content, viability and mitochondrial membrane potential of generated platelets were determined by Draq5, Calcein AM and TMRM staining, respectively.

To detect the viability and mitochondrial membrane potential of pulmonary endothelial cells (ECs), assays for Calcein Deep Red retention in ECs and TMRM accumulation in active mitochondria were conducted by FACS. Pulmonary ECs were isolated from perfused lungs under air- or pure-nitrogen- ventilation or without ventilation for ~2 h. In brief, lung was harvested and minced using scissors. The minced lung tissues were digested by collagenase I (3 mg/mL) in serum-free IMDM medium at 37 °C for 40 min, followed by filtration through a 70 μm strainer. Cells were then washed twice with serum-free IMDM medium and stained with FITC-conjugated anti-CD31/PECAM-1 (1:100 dilution), or anti-CD102/ICAM-2 antibodies (1:100 dilution), followed by loading with Calcein Deep Red or TMRM dyes for 25 min at room temperature. Pulmonary ECs from fresh lungs served as control.

Platelet surface glycoproteins were measured by incubating with FITC-conjugated anti-mouse CD61, CD42b, CD42d, CD49b, GPVI antibodies (all 1:100 dilution) or isotype-nonspecific IgG (1:100 dilution) for 20 min at room temperature before fixation.

To investigate the function of generated platelets (generated platelets derived from CD41-FITC stained MKs), assays for αIIbβ3 integrin activation (JON/A-binding) and P-selectin expression were performed. Washed platelets were stimulated with 2 U/mL thrombin or 5 μg/mL CRP-XL for 10 min followed by incubation with PE-JON/A or

PE-P selectin antibodies (both 1:100 dilution) for 20 min at room temperature before fixation. Tirofiban and PE-IgG were used to exclude nonspecific binding for the measurement of PE-conjugated JON/A or PE-conjugated P-selectin exposure, respectively. Resting platelets served as negative controls.

Samples were acquired and analysed on a BD Accuri™ C6 Plus flow cytometer using BD Accuri C6 Plus Software version 1.0.23.1.

## Two-photon imaging and platelet counting in the lung vasculature

After overnight fixation at 4 °C, the lung was staged on a MicroSlicer device (Type: DTK-1000N, UK) and cut into small slices with 800 μm thickness and flat surface. Lung slices from different lobes were fixed into a 100 mm cell culture dish and immersed in PBS for two-photon imaging.

Imaging was performed using a DeepSee multiphoton laser (Spectra Physics) attached to an upright SP8 confocal microscope (Leica Microsystems). All images were acquired using a 25×/0.95NA water dipping lens, with LAS X software (version 3.1.5.16308, Leica Microsystems). For CD41-FITC or CD41-PE, excitation was provided by tuning the multiphoton laser to 927 nm, for Hoechst 33342, excitation was provided by tuning the multiphoton laser to 750 nm. The resultant fluorescence for both scans passing off a SP500 dichroic beam splitter and through both a SP680 filter and either a 525/50 nm bandpass filter to selected only CD41-FITC signal, a 630/75 nm bandpass filter to select only CD41-PE signal or a 460/50 nm bandpass to select only Hoechst 33342 signal. CD41-FITC or CD41-PE fluorescence was detected using non-descanned Hybrid detectors (Leica Microsystems) and Hoechst 33342 fluorescence was detected using a non-descanned PMT. Images were acquired with an additional zoom of 2.5× with 1772.5 × 1772.5 pixels (XY), with an effective pixel size of 100 nm. Z stacks were captured with a z-step spacing of 2 μm. All images were capture using a scan speed of 400 Hz with a bidirectional scan.

To count generated platelets retained in the lung vasculature, image analysis was performed using Fiji ImageJ-win 64 software and ten z-stacks were analysed as a single volume in extended focus. The total volume of each analysed lung two-photon extended focus sample was therefore (177.25 × 177.25 × 2) × 10 = 628351.25 μm³, and the numbers of fluorescent events (~2–6 μm diameter) were counted manually. Total lung volume was determined by a fluid displacement approach, as per details in Supplementary Fig. 1f.

## Confocal microscope imaging

Confocal images were acquired on an inverted SP8 confocal microscope (Leica Microsystems) attached to a DMI6000 microscope frame (Leica Microsystems), using LAS X software (version 3.5.7.23225, Leica Microsystems). All images were obtained using a 100×/1.44 NA oil immersion objective. Excitation was provided by either a 405 nm laser (Hoechst 33342) or 488 nm laser (for CD41-FITC) or 561 nm laser (for CD41-PE) and the resultant fluorescence was detected using a Hybrid detector in the range 410–470 nm for Hoechst 33342, 495–570 nm for CD41-FITC or 571–650 nm for CD41-PE. A total of 50 randomly chosen fields of view were imaged and where z-stacks were acquired, a 2 μm z-step spacing was used for Supplementary Movies 7–10, or a 0.5 μm z-step spacing for Supplementary Movie 11.

To obtain an estimate of the dimensions of each nucleus in XY, z-stacks were loaded into Fiji imageJ and subjected to a maximum intensity projection and a region of interest (ROI) was manually drawn around an individual nucleus. The inbuilt measure function within Fiji imageJ was used. To obtain the dimensions of the major and minor axis, the 'fit ellipse' option was enabled. To estimate the nuclear depth, analysis was performed in Fiji ImageJ stacks were loaded, and a cell nucleus was selected using the ROI tool. The average intensity profile of this nucleus was plotted in z and then fitted with a gaussian profile using the built-in plot and fitting tools of Fiji imageJ. From the fitted

parameters the depth of the nucleus was estimated using the full width at half maximum (FWHM) of the gaussian fit. The FWHM was calculated as $2\sqrt{(2\ln 2)}\sigma$. This process was manually repeated for multiple cell nuclei.

## Transmission electron microscope imaging

To visualize and compare the ultrastructure of generated and control platelets by transmission electron microscopy (TEM), host platelets were first depleted by intraperitoneal administration of anti-GPIbα antibody R300 (2 μg/g bodyweight) prior to MKs infusion through the heart-lung preparation. After 18 passages, the generated platelets were pelleted by centrifuging the collected perfusate. The platelet pellet was then resuspended in a cacodylate-buffered glutaraldehyde fixative and fixed at 4–8 °C overnight and then post-fixed in osmium ferrocyanide. Fixed cells were then embedded in a solidifying BSA/glutaraldehyde gel. Gel-embedded platelets were stained with uranyl acetate and lead aspartate followed by dehydration with ethanol and infiltrated with Epon resin in a Tissue Processor (Leica EMTP). 70 nm sections were cut from blocks with a Reichert Ultracut E and imaged with a Tecnai 12 electron microscope (ThermoFisher UK).

## In vitro thrombus formation

In brief, ibidi μ-Slide VI 0.1 chips were coated with 50 μg/mL collagen overnight at 4 °C before being flushed and blocked with 2% fatty acid-free BSA prepared in HEPES-Tyrode's buffer. Freshly drawn mouse blood was collected from the inferior vena cava using 10 U/mL heparin and 20 μM D-phenylalanyl-prolyl-arginyl chloromethyl ketone (PPACK) as anticoagulant, following euthanasia by rising $CO_2$. Mouse blood was mixed with 2 mL NaCl (150 mM) and centrifuged to remove platelet rich plasma (PRP). MKs stained with 2 μM $DiOC_6$ were passaged through the mouse-lung preparation 18 times, and ~0.7 mL perfusate was mixed with 1.3 mL mouse blood lacking PRP and incubated with CellTracker™ Red CMTPX dye (1:1000 dilution) for 10 min.

The mixed sample was then transferred to a 5 mL syringe and perfused using an Aladdin AL-1000 syringe pump (World precision instruments, United Kingdom) through the ibidi slide, at a shear rate of 1000/s for 20 min. Platelets were fixed by perfusion of 4% paraformaldehyde through channels for 4 min before nonadherent cells were flushed away with HEPES-Tyrode buffer.

Thrombus formation was determined by generating confocal z-stacks (1024 × 1024 pixels, 0.787 μm z stack distance) from 5 randomly chosen fields of view using a Leica SP8 confocal microscope. Images were acquired using a 20×/0.7 NA air immersion objective, using LAS X software (version 3.5.7.23225, Leica Microsystems). Excitation was provided by either a 488 nm ($DiOC_6$) or 561 nm (Cell-Tracker™ Red CMTPX) laser with the resultant fluorescence being detected by Hybrid detectors in the range 498–551 nm ($DiOC_6$) or 571–623 nm (CellTracker Red CMTPX).

## Two-photon intravital microscopy of the bone marrow

Mice were anaesthetized by intraperitoneal injection of medetomidine 0.5 μg/g, midazolam 5 μg/g and fentanyl 0.05 μg/g body weight. A 1 cm incision was made along the midline to expose the frontoparietal skull, without damaging the bone tissue. To immobilize the head while imaging, the mouse was fastened with a stereotactic holder on a heated customized stage. Bone marrow vasculature was visualized by intravenous injection of anti-CD105-AlexaFluor 546 antibody (1 μg/g body weight) and 100 μL AlexaFluor 546-labelled BSA. Platelets and MKs were stained intravenously with anti-GPIX-AlexaFluor 488 antibody (1.5 μg/g body weight). Images and time-lapse videos were taken using a 20× water objective with a numerical aperture of 0.95 (Leica Microsystems) on a confocal TCS SP8 MP (Leica Microsystems) equipped with a tunable broadband laser (Coherent), with LAS X software (version 3.1.5.16308, Leica Microsystems).

## Image analysis and 3D segmentation

All image analysis was performed using Fiji ImageJ-win 64 software. 3D cell segmentation was performed in Fiji ImageJ using the Modular-ImageAnalysis (MIA) workflow automation plugin[55,56].

## Statistical information

All statistical analysis was performed on data from a minimum of 3 independent experimental repeats. Graphs were prepared using GraphPad Prism 9 software (GraphPad Software Inc., San Diego, CA, USA). Quantified data are presented as mean value ± the standard error of the mean (SEM) for each experimental condition. A value of $p < 0.05$ was considered statistically significant and determined using either two-tailed unpaired t test for normally distributed data (comparison of two groups) or two-tailed Mann–Whitney test for non-normally distributed data (comparison of two groups) or two-way ANOVA with Tukey's or Šídák's multiple comparisons test, as indicated in figure legends. Choice of test was determined by assessment of normality of data, and whether single or multiple comparison was required.

## Reporting summary

Further information on research design is available in the Nature Portfolio Reporting Summary linked to this article.

## Data availability

The datasets generated during and/or analysed during the current study are included as Source Data. Source data are provided with this paper.

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

## Acknowledgements

We gratefully acknowledge the Wolfson Bioimaging Facility for their support and assistance in this work. We are also grateful to Professor Jack Mellor, University of Bristol, for the use of the tissue slicer machine. This work was supported by a Wellcome Trust Investigator Award to A.W.P. (219472/Z/19/Z) and C.G. (219472/A/19/Z) and grants from the British Heart Foundation (RG/15/16/31758 to A.W.P., SP/F/21/150023 to A.W.P., C.G. & I.H. and PG/16/102/32647 to A.W.P. and E.O.A.). B.N. was funded by Deutsche Forschungsgemeinschaft (DFG, German Research Foundation, project number 374031971- TRR 240/project A01).

## Author contributions

Conceptualization: X.Z., C.G. and A.W.P.; Methodology: X.Z., D.A., T.G.W., N.T., M.E., S.Z.B., Y.L., C.R.N., P.B. and S.J.C.; Visualization: X.Z., D.A., M.E., P.B. and C.R.N.; Resources: D.A., E.W.A., A.K.W. and J.B.B.; Writing—original draft: X.Z., D.A., M.E., S.Z.B., C.R.N., S.J.C. and A.W.P.; Writing— review & editing: X.Z., D.A., C.M.W., P.W.G., E.C.H., B.N., I.H., C.G. and A.W.P.; Supervision & Investigation: X.Z. and A.W.P.; Funding acquisition: E.O.A., C.G., I.H., B.N. and A.W.P.; Project administration: A.W.P.

## Competing interests

P.W.G. and E.C.H. receive funding from TroBio Therapeutics, a company commercialising tropomyosin-targeting drugs. P.W.G. and E.C.H. are directors and shareholders of TroBio. All other authors declare no competing interests.
