## [Peer Review File · Nature Communications]

REVIEWER COMMENTS

Reviewer #1 (Remarks to the Author):

In this manuscript, Zhao and colleagues show that the pulmonary vasculature plays a key role in blood platelet release. They developed a technologically challenging approach using an ex vivo mouse heart-lung model. They perfused pre-stained mouse MKs obtained from in vitro culture and found that multiple passages through the lungs were required to maximally fragment MKs. In parallel, they recovered particles in the perfusate that they characterized as young platelets, capable of being activated and incorporated into a thrombus in vitro. They also developed an in vitro chamber mimicking the microvasculature of the lungs that effectively releases platelets, but nevertheless to a lesser extent than the lungs. They demonstrated that ventilation and oxygenation of the lungs were essential for maximal MK fragmentation. Finally, they found that tropomyosin 4 was essential for the final stages of platelet formation in the lungs.

The manuscript is really interesting and very well written. This is the first study to show that a platelet count equivalent to the theoretical calculation can be obtained in vitro from MK grown in culture. The novelty clearly lies in the use of the heart-lung model, allowing for optimal platelet release from cultured MKs and the study of a new platelet release pathway. Nevertheless, I have a few points that I would like the authors to clarify.

Major points.

1. What is the size range of the perfused MKs?
2. Figure 1e is not clear to me. The histogram on the left shows the P1 gate, and there is a large proportion of CD41-negative cells while the "generated platelets" are only a small fraction. Since the lungs are washed prior to infusion of labeled MKs, what is this CD41-negative population with the size of platelets? Please explain.
3. Figure 1h and 1j: the authors should show the statistics.
4. The idea that lung ECs contribute to platelet release is really interesting. The authors should detail in Materials and Methods how they isolate lung ECs. To take this a step further, what happens if MKs are perfused into the heart-lung model that has been previously fixed with PFA and flushed? Can platelets be generated by a passive endothelium?
5. Also, if the microfluidic chamber is kept under N₂ during perfusion, does this change the final platelet release ratio?
6. Figure 4a: Please show the non-activated control platelets to see the baseline for Jon/A-PE and CD62P labeling. If the mean \pm SEM, as indicated in the legend, is reported here, then the experiment is very heterogeneous, especially with the generated platelets. The authors should also show the individual points and discuss why some batches of generated platelets are less able to be activated. In Fig. 3b,

there appear to be 2 populations of generated platelets with different sizes rather than a continuum, while still being different from native platelets. Do these 2 populations respond differently to agonist stimulation?

7. Figure 4d. It would be useful to also have the bright field to visualize platelet thrombi. In the text, the authors mention that the generated platelets are "white", whereas in the fluorescent image, the blue is visible but not the white. Please make it clearer.

8. Figure 5a: The authors mention the presence of a naked nucleus. It could be very interesting to verify whether this is really a naked nucleus or a nucleus surrounded by a plasma membrane. In erythrocytes, for example, the nucleus is lost when it is extruded as a pyrenocyte, a nucleus surrounded by a plasma membrane that exposes phosphatidylserine on its surface to attract macrophages. This can be assessed by non-specifically labeling the plasma membrane or by performing TEM.

9. Throughout the discussion, the authors suggest that under normal conditions, whole MKs enter the lungs to release platelets, and that lung defects prevent MKs from being fragmented into platelets, resulting in higher numbers of MKs in patients with lung disease. What is the evidence for significant transmigration of whole MKs under homeostatic conditions? Have the authors quantified the fraction of MK nuclei in the mouse lungs versus MKs in the marrow? The authors should consider that the passage of whole MKs into the circulation might rather be a rare event under physiological conditions, which is greatly increased during inflammatory reactions, especially as reported during a viral infection such as Covid19, explaining the higher amount of MKs in the lungs (and other organs). This does not question the fact that the lungs are key to the release of platelets from MK fragments that enter the marrow vasculature.

Minor points.

Page1 line 65, from the recent literature it seems that contrary to what is written, platelets produced in vitro are now able to satisfactorily respond to agonists.

The scale of the bar in Supplemental Figure 1 is lacking

The schema of the microfluidic chamber (right panel) is not so clear. Could you indicate the flow direction and provide also a photomicrograph?

Figure 6C: Please use the same scale for the graph as the one in Fig. 5b to help compare the bar size.

Statistics: why use paired t-test? some data clearly do not present a normal distribution, hence the tests should be adjusted accordingly.

Reviewer #2 (Remarks to the Author):

The focus of this manuscript by Zhao and colleagues is on mechanisms by which megakaryocytes release platelets in the lung vasculature. There is ample indirect and direct in vivo evidence that the lung plays a major role in thrombopoiesis, but the mechanisms involved have not been fully elucidated. The authors use an ex vivo lung-heart block to repeatedly infuse megakaryocytes through the lung vasculature and they observed that megakaryocytes can repeatedly passage through the lung with increasing platelet production and progressive enucleation. The final component of this process is dependent on the actin regulator, TPM4.

Major criticisms:

1. Some references are missing on the role of the lung in platelet production, such as those by the Poncz group where they injected megakaryocytes intravenously and showed that the lung is capable of functional platelet production (PMID: 20972336; PMID: 25852052).
2. More detail is needed on the lung-heart block set-up. For example, with the ventilation, was end-expiratory pressure applied, like is conventionally done in the mechanical ventilation of mouse or human lungs? If not, this could promote end-expiratory lung collapse and shunting. Megakaryocytes were “pumped into the right ventricle” per the Methods section. Was this just a push of the syringe or were they placed into the perfusion pump and infused at 0.35 ml/min per the Methods? How was this infusion rate selected and is it physiologic? This is relevant regarding perfusion pressures and sheer stress in the lung microvasculature.
3. Where is the evidence for the statement that “MKs substantially, rapidly and reversibly, deform their shape to passage through the capillary network of the lung.” Direct imaging evidence is needed to support this statement as it does stretch credulity that MKs of up to 100 um in size could squeeze through a lung capillary segment that is on average 5 um in diameter and sometimes smaller.
4. Overall, the study has merit for advancing methodology for the in vitro production of platelets but in terms of advancing concepts of homeostatic, in vivo platelet production in the lung, the limitations of the ex vivo heart-lung perfusion setup with the lack of a systemic circulation are problematic.

Reviewer #3 (Remarks to the Author):

There has been a debate over the last 130 years as to where the tiny (3 μ diameter) anucleate platelets are released from very large (100 μ diameter) multinucleated megakaryocytes, a process called thrombopoiesis. The majority camp says that this occurs in the bone marrow venous sinuses, but a persistent minority says it is also (or mostly) the lungs. This argument is of importance for reasons including strategies applicable to providing non-donor-derived platelet transfusions.

This manuscript by Xiaojuan Zhao, et al., "Highly efficient platelet generation in lung vasculature reproduced by microfluidics" focuses on a technical tour-de-force mouse model of repeatedly reinfused murine megakaryocytes into a heart-lung controlled model, resulting in the release of a large number of platelets. The model provides further support to the in situ pulmonary observations made by E Lefrançois in Nature that bone marrow-derived megakaryocytes migrate to the lung to release platelets. Moreover, if correct, it also provides two new important insights into this process: 1) thrombopoiesis is not a one-and-done process, but requires multiple recycling of the processed megakaryocytes to release all of the platelets. 2) That the pulmonary bed may be unique in this process because of air exchange and the endothelial lining and the process of ventilation.

There are significant issues with this paper though:

1. The model is under-detailed for a full critical review.

Critical details as to what is left circulating after the mouse is prepped and their blood is washed out is missing and should be included in a Supplement.

What are the blood counts in the animals under study as part of the observed outcome may be due to severe anemia and hypoxic injury within and without the lung-heart model. Is there DIC for example. Was it measured?

Is the heart still functional and beating or is blood flow through the lungs all regulated by an artificial pump? One would expect a nonfunctioning heart to develop intramural thrombi.

How long is each experiment if you recycle the blood 18 times? What are the details of each recycle? Is the reperfused sample hypoxic during this process or reoxygenated.

In some experiment an anti-platelet antibody was given, presumably in addition to the blood washout to decrease native platelet levels? Why this added step and what were the final platelet counts with and without this treatment? Wouldn't the antibody also bind to the infused megakaryocytes and the generated platelets?

2. There are other biological questions about the data and its interpretation. The anti-CD41 antibody should label the alpha granules as well as the megakaryocyte surface but that was not apparent in the images. No granules were noted. Furthermore, if this is an intact anti-CD41 antibody, it may also activate the megakaryocytes and generated platelets and often a F(ab)₂ fragment needs to be used instead in in vivo mouse models.

It is also concerning that under N₂ that the pulmonary endothelium was uninjured and still functional. Markers of apoptosis should have been investigated compared to controls in the various stages and should have shown injury at least under the marked hypoxic condition of N₂ ventilation for a significant time window. The data saying that the endothelium was healthy is thus a concern and also goes against the video of the lungs in the N₂ mouse, which clearly shows a large thrombus.

3. The videos are technically problematic. There are a number of mislabeled videos, especially internal within the videos. Most of these videos are highly pixelated, making data interpretation difficult and the two-photon microscopy do not contribute to better-quality prior videos by Junt and Massberg.

4. Platelet-generating device in Fig. 3C platelet is noncontributory to the bottomline message nor is it well-developed to explain how it is informed by the heart-lung model or tests distinct aspects of the pulmonary bed (especially oxygenation and ventilation. At a minimum it should be in the supplement and a photograph of the device and a high-quality video of thrombopoiesis are needed. Finally having shown that one can infuse MKs into recipients and have functional platelets released, the authors then punt to a device for clinical platelet transfusion instead of completing the circle of their thought and suggest that infused MKs may be an alternative therapy.

5. The presentation is repetitive at points and often imprecise. The discussion is too long and cumbersome.

An annotated PDF will hopefully be returned with additional smaller issues noted.

Morty POncz

REVIEWER COMMENTS

Reviewer #1 (Remarks to the Author):

In this manuscript, Zhao and colleagues show that the pulmonary vasculature plays a key role in blood platelet release. They developed a technologically challenging approach using an ex vivo mouse heart-lung model. They perfused pre-stained mouse MKs obtained from in vitro culture and found that multiple passages through the lungs were required to maximally fragment MKs. In parallel, they recovered particles in the perfusate that they characterized as young platelets, capable of being activated and incorporated into a thrombus in vitro. They also developed an in vitro chamber mimicking the microvasculature of the lungs that effectively releases platelets, but nevertheless to a lesser extent than the lungs. They demonstrated that ventilation and oxygenation of the lungs were essential for maximal MK fragmentation. Finally, they found that tropomyosin 4 was essential for the final stages of platelet formation in the lungs. The manuscript is really interesting and very well written. This is the first study to show that a platelet count equivalent to the theoretical calculation can be obtained in vitro from MK grown in culture. The novelty clearly lies in the use of the heart-lung model, allowing for optimal platelet release from cultured MKs and the study of a new platelet release pathway. Nevertheless, I have a few points that I would like the authors to clarify.

Major points.

1. What is the size range of the perfused MKs?

We have now included this size range of the perfused MKs in Supplementary Fig. 1c, with a range of 12.7 - 79.2 μm .

2. Figure 1e is not clear to me. The histogram on the left shows the P1 gate, and there is a large proportion of CD41-negative cells while the "generated platelets" are only a small fraction. Since the lungs are washed prior to infusion of labeled MKs, what is this CD41-negative population with the size of platelets?

Thank you for raising this. These additional cells are likely to be stem cells, which are also small in size and often run within the same gate as platelets. We have now expanded Fig. 1d to include data from 0 passages, i.e, just prior to running cells through the lung-heart preparation for the first time. In Fig. 1e legend, we also now clarify that the CD41-negative populations are ones that include stem cells and host-derived platelets.

3. Figure 1h and 1j: the authors should show the statistics.

In Fig. 2a (original Fig. 1h) and its legend, we have now shown the statistics, which was performed using two-way ANOVA with Tukey's multiple comparisons. In Fig. 2c (original Fig. 1j) however, the data are not actually to be compared, because we were not looking to assess the relative contribution of platelets in the perfusate versus those retained in the lung vasculature. These are absolute numbers that, when added together, provide the total number of platelets generated per megakaryocyte, and so it is not appropriate to conduct any comparison for the data in Fig. 2c (original Fig. 1j).

4. The idea that lung ECs contribute to platelet release is really interesting. The authors should detail in Materials and Methods how they isolate lung ECs.

The method of how to isolate lung ECs is now described in the “Flow cytometry” part of the Methods section.

To take this a step further, what happens if MKs are perfused into the heart-lung model that has been previously fixed with PFA and flushed? Can platelets be generated by a passive endothelium?

We thank the reviewer for this suggested experiment, which is certainly of interest. We have tried to perform it several times, but experimentally it is difficult. The reason is that after perfusion of the tissue with PFA for 10 mins, its compliance and elasticity is markedly reduced. This meant that when we tried cannulating the right ventricle/pulmonary artery, the seal around the influx cannula was poor. In non-fixed tissue, we get a good seal because the tissue effectively closes up around the needle/cannula site (see comparison figures below), forming an effective seal. In the PFA-fixed tissue this was not the case, and there was substantial leakage of medium when fluid was perfused in. We could not therefore effectively generate enough pressure to allow perfusion of the vasculature for the experiment.

5. Also, if the microfluidic chamber is kept under N₂ during perfusion, does this change the final platelet release ratio?

Thank you for this question, which is an interesting one. We have gone back to conduct the generation experiment in the chamber under N₂, as suggested, and the new data are now shown in Fig. 3e. Importantly, the generation was reduced to almost zero, as in the lung system, indicating more clearly the critical role of oxygenation of megakaryocytes, specifically, in platelet generation. We have therefore included additional text in Results section.

6. Figure 4a: Please show the non-activated control platelets to see the baseline for Jon/A-PE and CD62P labeling.

We now show the levels of integrin activation and P-selectin expression on resting platelets in Fig 5a.

If the mean \pm SEM, as indicated in the legend, is reported here, then the experiment is very heterogeneous, especially with the generated platelets. The authors should also show the individual points and discuss why some batches of generated platelets are less able to be activated.

In the original submission we had shown S.D. rather than S.E.M. We have now changed the graph to show S.E.M. and have included individual datapoints as requested.

In Fig. 3b, there appear to be 2 populations of generated platelets with different sizes rather than a continuum, while still being different from native platelets. Do these 2 populations respond differently to agonist stimulation?

We agree with the reviewer that there are two apparent subpopulations of generated platelets, according to their size, and we have now explained this in the legend of Fig. 4b.

The subpopulation with the larger size were more responsive, by comparison with the smaller subpopulation, to thrombin and CRP in both integrin α IIb β 3 activation and P-selectin expression. These data are now shown in a modified Fig. 5a.

7. Figure 4d. It would be useful to also have the bright field to visualize platelet thrombi.

We agree and have now included a brightfield image from a middle plane of the thrombus, in the righthand panel of Fig. 5c.

In the text, the authors mention that the generated platelets are "white", whereas in the fluorescent image, the blue is visible but not the white. Please make it clearer.

We agree and have now corrected it by changing "white" to "blue".

8. Figure 5a: The authors mention the presence of a naked nucleus. It could be very interesting to verify whether this is really a naked nucleus or a nucleus surrounded by a plasma membrane. In erythrocytes, for example, the nucleus is lost when it is extruded as a pyrenocyte, a nucleus surrounded by a plasma membrane that exposes phosphatidylserine on its surface to attract macrophages. This can be assessed by non-specifically labeling the plasma membrane or by performing TEM.

We agree with the analysis of the reviewer. We find both situations and have expanded Fig. 6a (original Fig. 5a) and Supplementary Movie 9 where we had already shown images of nuclei that are genuinely naked, but now include a representative image of ones that are surrounded by thin/patchy CD41-stained membrane. In fact, the nucleus shown in lefthand panel also shows a thin staining of CD41 around the nucleus.

9. Throughout the discussion, the authors suggest that under normal conditions, whole MKs enter the lungs to release platelets, and that lung defects prevent MKs from being fragmented into platelets, resulting in higher numbers of MKs in patients with lung disease.

The literature to which we referred (Ref.30-33) showed variable platelet counts in individuals with lung pathologies or individuals living at high altitude. In other words that there is no consistent effect on platelet number by these conditions, and that the explanation for this is likely to be the complexity of the platelet generation process, in lung and marrow. In lung vasculature, we had described 4 factors which may influence the ability of MKs to generate platelets: oxygenation, ventilation, endothelial cells and structure of the lung microvasculature. Changes in one of these factors may therefore be compensated for by changes in other factors.

What is the evidence for significant transmigration of whole MKs under homeostatic conditions?
We agree that it will be important to address this question in a future study, because clearly we would like to know the relative significance of platelet generation in the lung versus the bone marrow, in physiological conditions, in humans *in vivo*. For this question in mouse, this is more clearly defined, for example in Junt et al. (Ref.29) and Fig. 8, where it has been easier to identify whole MKs in the circulation.

Have the authors quantified the fraction of MK nuclei in the mouse lungs versus MKs in the marrow? The authors should consider that the passage of whole MKs into the circulation might rather be a rare event under physiological conditions, which is greatly increased during inflammatory reactions, especially as reported during a viral infection such as Covid19, explaining the higher amount of MKs in the lungs (and other organs). This does not question the fact that the lungs are key to the release of platelets from MK fragments that enter the marrow vasculature.
We have not determined the fraction of MK nuclei in the mouse lung versus the marrow, mainly because our study addresses an *ex vivo* and *in vitro* approach to generate platelets, but also because in the *in vivo* setting these MK nuclei are likely to be removed rapidly by tissue macrophages, making this analysis difficult to interpret.

Minor points.

Page1 line 65, from the recent literature it seems that contrary to what is written, platelets produced *in vitro* are now able to satisfactorily respond to agonists.

The problem in the field generally now is to generate large numbers of platelets per MK, that are also functional. In fact, all current publications cite numbers of platelets generated per MK as being very low, and all under 100 platelets per MK. A smaller number of these publications indicate functionality, but although these may show functionality, this is associated with a very low production efficiency (platelets per MK). So, we have rephrased this line in the Abstract and Introduction to reflect this.

The scale of the bar in Supplemental Figure 1 is lacking.

We have now added the description of the scale bar to the legend of Supplement Fig 1d.

The schema of the microfluidic chamber (right panel) is not so clear. Could you indicate the flow direction and also provide a photomicrograph?

Thank you, and these have now been provided in Fig. 3c.

Figure 5b: Please use the same scale for the graph as the one in Fig. 5b to help compare the bar size.

Thank you, and we have now amended all the scale bars throughout the manuscript to make them all comparable.

Statistics: why use paired t-test? some data clearly do not present a normal distribution; hence the tests should be adjusted accordingly.

We agree that, the data shown for platelet size in Fig. 4b for generated platelets, the distribution is not Gaussian. We have therefore ensured that we have applied normality testing to all datasets in the manuscript, and have applied appropriate tests, as described now in the revised Methods: A value of $p < 0.05$ was considered statistically significant and was determined using either unpaired t-test for normally distributed data (comparison of two groups) or Mann-Whitney U test for non-normally distributed data (comparison of two groups) or two-way ANOVA with Tukey's multiple comparisons test, as indicated in figure legends. Choice of test was determined by assessment of normality of data (Kolmogorov-Smirnov analysis), and whether single or multiple comparison was required. We have included details of the test used in the individual figure legends.

Reviewer #2 (Remarks to the Author):

The focus of this manuscript by Zhao and colleagues is on mechanisms by which megakaryocytes release platelets in the lung vasculature. There is ample indirect and direct *in vivo* evidence that the lung plays a major role in thrombopoiesis, but the mechanisms involved have not been fully elucidated. The authors use an *ex vivo* lung-heart block to repeatedly infuse megakaryocytes through the lung vasculature and they observed that megakaryocytes can repeatedly passage through the lung with increasing platelet production and progressive enucleation. The final component of this process is dependent on the actin regulator, TPM4.

Major criticisms:

1. Some references are missing on the role of the lung in platelet production, such as those by the Poncz group where they injected megakaryocytes intravenously and showed that the lung is capable of functional platelet production (PMID: 20972336; PMID: 25852052).

We now have included these two references (Ref 14 -15) in Introduction section.

2. More detail is needed on the lung-heart block set-up. For example, with the ventilation, was end-expiratory pressure applied, like is conventionally done in the mechanical ventilation of mouse or human lungs? If not, this could promote end-expiratory lung collapse and shunting.

We have now added this additional detail to the Methods section on mouse *ex vivo* lung-heart preparation. To avoid end-expiratory lung collapse, as indicated in Fig 1a, the connector for expiration air was connected to a Gottlieb valve (tube immersed in water; immersion 2-3 cm depth to induce a positive end-expiratory pressure).

Megakaryocytes were "pumped into the right ventricle" per the Methods section. Was this just a push of the syringe or were they placed into the perfusion pump and infused at 0.35 ml/min per the Methods?

This has been correctly described now in the Methods section also, on mouse *ex vivo* lung-heart preparation.

How was this infusion rate selected and is it physiologic? This is relevant regarding perfusion pressures and shear stress in the lung microvasculature.

The flow rate we used in our experiments achieves a pressure of 2.1–3.5 mmHg, as measured by a pressure transducer as shown in Fig. 1a. This is substantially lower than physiological pressures in the mouse pulmonary artery, which has been estimated to be 25 mmHg. We have added this to the text of the Methods section in the manuscript.

3. Where is the evidence for the statement that “MKs substantially, rapidly and reversibly, deform their shape to passage through the capillary network of the lung.” Direct imaging evidence is needed to support this statement as it does stretch credulity that MKs of up to 100 μm in size could squeeze through a lung capillary segment that is on average 5 μm in diameter and sometimes smaller.

Our Fig. 1b and 1c data show that we can recover intact MKs in perfusates of lung vasculature *ex vivo* and *in vivo*. We have interpreted this as meaning that these large cells deform their shape to progress through the microvasculature, and indeed *in vivo* imaging of MKs in bone marrow sinusoids, for example, indicates that they are capable of substantial change in shape and appear in the vasculature (reference to Brown et al⁶, and Junt et al²⁹ and to intravital imaging in bone marrow in Fig. 8 and Supplementary 13-14). However, although we cannot think of alternative explanations for interpretation of these data, we agree that we lack direct imaging data here, which would form a major new piece of work. For this reason, we have removed this statement from the manuscript.

4. Overall, the study has merit for advancing methodology for the *in vitro* production of platelets but in terms of advancing concepts of homeostatic, *in vivo* platelet production in the lung, the limitations of the *ex vivo* heart-lung perfusion setup with the lack of a systemic circulation are problematic.

We agree that this work advances our understanding and development of systems to generate platelets *in vitro*, and also agree that it will be important now to determine the relative role of pulmonary intravascular platelet generation in the *in vivo* setting, both in the mouse model system but also importantly in the human. However, the merits of the *ex vivo* system we have developed here, and the *in vitro* microfluidic model, have allowed us to understand details of platelet generation that had previously not been known, and also will serve as a platform for further discovery science into the molecular and cellular processes, as well as potentially for development of larger scale platelet generation systems with human cells. These are all future developments that can springboard from the work presented in this manuscript.

Reviewer #3 (Remarks to the Author):

There has been a debate over the last 130 years as to where the tiny (3 μ diameter) anucleate platelets are released from very large (100 μ diameter) multinucleated megakaryocytes, a process called thrombopoiesis. The majority camp says that this occurs in the bone marrow venous sinuses, but a persistent minority says it is also (or mostly) the lungs. This argument is of importance for reasons including strategies applicable to providing non-donor-derived platelet transfusions.

This manuscript by Xiaojuan Zhao, et al., “Highly efficient platelet generation in lung vasculature reproduced by microfluidics” focuses on a technical tour-de-force mouse model of repeatedly reinfused murine megakaryocytes into a heart-lung controlled model, resulting in the release of a large number of platelets. The model provides further support to the *in situ* pulmonary observations made by E Lefrançois in Nature that bone marrow-derived megakaryocytes migrate to the lung to release platelets. Moreover, if correct, it also provides two new important insights into this process: 1) thrombopoiesis is not a one-and-done process, but requires multiple recycling of the processed megakaryocytes to release all of the platelets. 2) That the pulmonary bed may be unique in this process because of air exchange and the endothelial lining and the process of ventilation. There are significant issues with this paper though:

1. The model is under-detailed for a full critical review.

Critical details as to what is left circulating after the mouse is prepped and their blood is washed out is missing and should be included in a Supplement.

What are the blood counts in the animals under study as part of the observed outcome may be due to severe anemia and hypoxic injury within and without the lung-heart model. Is there DIC for example. Was it measured?

Is the heart still functional and beating or is blood flow through the lungs all regulated by an artificial pump? One would expect a nonfunctioning heart to develop intramural thrombi.

We thank the reviewer for these comments. All the experiments in the manuscript are based on an *ex vivo* model, not an *in vivo* one, where the animal is first sacrificed before the heart-lung vasculature is isolated *in situ* and perfused.

So, the experimental details are described for this *ex vivo* model, and for this reason there is no remaining circulation because the heart is no longer pumping. Because of this, there are also no associated blood counts for these preparations, and there are no intramural thrombi in the heart as well. This is because the fluid we perfuse through the system is not blood but rather megakaryocytes in suspension in cell culture medium. Also, prior to the perfusion we flush the vasculature with heparinized Krebs-HEPES buffer (detailed in the Methods section), therefore the chances of any clotting in the system are minimized.

We have provided additional details of our experimental approach, in Fig. 1a and in the Methods section. These details include how we ensure that the lung does not collapse by including end-expiratory positive pressure. We have also included measurement of perfusion pressures to ensure that this is not so high that it would damage pulmonary vasculature or structure.

How long is each experiment if you recycle the blood 18 times? What are the details of each recycle? Is the reperfused sample hypoxic during this process or reoxygenated.

For these experiments we suspend megakaryocytes in 2mLs medium, and perfuse these cells through the lung vasculature at 0.35 mL/min. Each passage through the lung-heart model therefore takes approximately 5-6 mins. The full time for 18 passages therefore takes approximately 2 hours. This point has been highlighted in the Results section of the text. During all this time, as samples perfuse, the lung is artificially ventilated with air (or in some experiments not ventilated, or ventilated with pure N₂, as described).

In some experiment an anti-platelet antibody was given, presumably in addition to the blood washout to decrease native platelet levels? Why this added step and what were the final platelet

counts with and without this treatment? Wouldn't the antibody also bind to the infused megakaryocytes and the generated platelets?

Here we use antibodies to label infused megakaryocytes, prior to perfusion through the lung vasculature, in our *ex vivo* lung-heart preparation. So, the antibodies are not used to decrease any native platelet levels, but simply to mark the megakaryocytes, and platelets generated from them, so that we may be able to recognize them subsequently by FACS or imaging approaches.

2. There are other biological questions about the data and its interpretation. The anti-CD41 antibody should label the alpha granules as well as the megakaryocyte surface but that was not apparent in the images. No granules were noted.

We labelled the megakaryocytes (MKs) with anti-CD41 antibodies, so as to mark the membrane of the MKs. We found that incubation of MKs with labelled anti-CD41 for 3 hours marks both the surface plasma membrane, but also allows time for the antibody to penetrate through surface-connected canalicular system to the internal membrane of the demarcation membrane system (DMS), which acts the reservoir for membranes for the generated platelets. So, since there is no permeabilization of the MKs, it is not possible for the labelled antibody to reach the alpha-granules, and this is why we do not see granule staining, using this protocol.

Furthermore, if this is an intact anti-CD41 antibody, it may also activate the megakaryocytes and generated platelets and often a F(ab)₂ fragment needs to be used instead in *in vivo* mouse models. In this study we have only used mouse MKs, which lack expression of Fc receptor FcγRIIA, unlike their human counterpart. So, for the mouse system in *ex vivo*, we are able to use intact antibodies without the possibility of activation of the MKs or their derived platelets.

It is also concerning that under N₂ that the pulmonary endothelium was uninjured and still functional. Markers of apoptosis should have been investigated compared to controls in the various stages and should have shown injury at least under the marked hypoxic condition of N₂ ventilation for a significant time window. The data saying that the endothelium was healthy is thus a concern and also goes against the video of the lungs in the N₂ mouse, which clearly shows a large thrombus. We thank the reviewer for these comments. All the experiments in the manuscript are based on an *ex vivo* model, not an *in vivo* one.

In Fig. 3a and b, we address the health status of the endothelial cells in the system upon exposure to N₂. We showed that their viability, as determined by retention of the vital dye Calcein Deep Red, was normal. However, the cells are clearly under some degree of stress, because the mitochondrial membrane potential, as measured by the fluorescence dye tetramethyl rhodamine methyl ester (TMRM), was decreased by comparison with air-ventilated lung-derived endothelial cells.

TMRM (and related molecules, such as Tetramethylrhodamine, ethyl ester (TMRE)) is used as a dye to distinguish dead and apoptotic cells from live and non-apoptotic cells (e.g. Barteneva et al. 2014, J Histochem Cytochem. 62(4): 265–275), and so the majority of cells in our analysis have been shown to be viable and non-apoptotic.

The large fluorescent objects that can be seen in lung structure in N₂-ventilated lungs are not thrombi, but rather are clusters of MKs.

3. The videos are technically problematic. There are a number of mislabeled videos, especially internal within the videos. Most of these videos are highly pixelated, making data interpretation

difficult and the two-photon microscopy does not contribute to better-quality prior videos by Junt and Massberg.

We have now addressed the issues with the videos, which are now saved in a different file format so as not to lose resolution or pixel density. The labelling is now correct for all videos.

The paper by Junt et al, 2007, shows imaging of MKs in mouse marrow in living tissue. This differs from the model we have established, which is *ex vivo* (not *in vivo*) in mouse pulmonary vasculature. We have used two-photon microscopy to image lung after perfusion of MKs into the pulmonary vasculature, to determine numbers of MKs lodged in that vasculature and generated platelets retained there. This imaging is of fixed lung tissue, not live cell imaging.

4. Platelet-generating device in Fig. 3C platelet is noncontributory to the bottom line message nor is it well-developed to explain how it is informed by the heart-lung model or tests distinct aspects of the pulmonary bed (especially oxygenation and ventilation. At a minimum it should be in the supplement and a photograph of the device and a high-quality video of thrombopoiesis are needed. Finally having shown that one can infuse MKs into recipients and have functional platelets released, the authors then punt to a device for clinical platelet transfusion instead of completing the circle of their thought and suggest that infused MKs may be an alternative therapy.

This microfluidic chamber was used to mimic the flow of MKs in the capillary bed of the pulmonary vasculature. In this way, particularly because there is no mechanical ventilation or endothelium in this system, was helpful to determine the relative contribution of these elements in platelet generation. The chambers are made from PDMS, which is a gas-permeable structure, and has also enabled us to address the role of oxygenation, in isolation from other factors, as now shown in the new Fig. 3e where N₂ replacement of air ablates platelet generation. We have also provided a further diagram of the details of the microfluidic device, plus a photomicrograph (Fig. 3c).

5. The presentation is repetitive at points and often imprecise. The discussion is too long and cumbersome.

The work presented has been complexed to perform and therefore details were needed in the descriptions. The data show several novel findings, including demonstration of passage of MKs through pulmonary vasculature, enucleation of MKs as part of the platelet generation process, a critical role for oxygenation shown in both the *ex vivo* lung preparation and the microfluidic chamber and the generation of very large numbers of platelets from MKs in vitro. These novel findings needed detailed presentation and discussion, and the text falls within the word limit for the journal.

REVIEWER COMMENTS

Reviewer #1 (Remarks to the Author):

I thank the authors who responded satisfactorily to all my comments and the manuscript is acceptable for publication

Reviewer #2 (Remarks to the Author):

Thank you for the responses. The details of the heart-lung prep are now much clearer. I have one remaining question. Did the application of the end-expiration device (Gottlieb valve) affect any of the results such as those in Figure 1b or Figure 2a? If possible, data should be shown with and without the device.

Reviewer #3 (Remarks to the Author):

This revised manuscript by Zhao, X, et al “Highly efficient platelet generation in lung vasculature reproduced by microfluidics” is an innovative examination of the process by which the lungs release platelets from megakaryocytes that should be impactful to readers interested in hematopoiesis and platelet biology and may also have impact on clinical platelet transfusions. The manuscript focuses on two complementary systems, a mouse heart-lung model where efflux containing megakaryocytes are reinfused up to 18 times and a microfluidic device to try to simulate the lungs.

Overall, the demonstration that megakaryocytes require multiple passage through the lungs to release platelets is convincing as are the studies on the subsequent processing of the cytoplasmic fragments to functional platelet-like particles.

There were a few concerns that need to be addressed in these studies:

1) In Fig 1d, why is that P1 window more “plt-like” at 0 passages and the P2 larger after 18 passages. Seems the reverse of what to expect.

2) In Fig 1f, annexin v or p-selectin or Jon A binding would have been useful and more commonly done to show that you were generating “happy” platelets.

3) While EC injury/death by inhalation of nitrogen or no ventilation is mentioned, it needs to be given more equal time as a possible explanation in the Results and Discussion for why platelet formation was not seen. Staining for surface markers compatible with endothelial injury like loss of surface thrombomodulin or extrusion of VWF would have been important.

The microfluidic studies are complementary but not as well developed.

1) The similarity between the microfluidic design and lung vasculature is not demonstrated and the discussion should be altered to reflect that especially as no other design was studied.

2) The device has no endothelial lining and that limitation and its implications discussed.

3) Fig 3 was shown with no mouse platelet control and that would have been an important comparative. Again, markers of activated platelets should have been measured.

The manuscript with additional comments is hopefully also returned.

**Highly efficient platelet generation in lung vasculature**
**reproduced by microfluidics**

Xiaojuan Zhao^{1*}, Dominic Alibhai², Tony G. Walsh¹, Nathalie Tarassova¹, Maximilian
Englert³, Semra Z. Birol¹, Yong Li¹, Christopher M. Williams¹, Chris R. Neal², Philipp
Burkard³, **Stephen J. Cross²**, Elizabeth W. Aitken¹, Amie K. Waller⁴, Jose Ballester-Beltran⁴,
Peter W. Gunning⁵, Edna C. Hardeman⁵, Ejaiife O. Agbani⁶, Bernhard Nieswandt³, Ingeborg
Hers¹, Cedric Ghevaert⁴ & Alastair W. Poole^{1*}

¹School of Physiology, Pharmacology and Neuroscience, Biomedical Sciences Building,
University of Bristol, Bristol, BS8 1TD, UK.

²Wolfson Bioimaging Facility, Biomedical Sciences Building, University of Bristol, Bristol,
BS8 1TD, UK.

³Institute of Experimental Biomedicine, University Hospital Würzburg, and Rudolf Virchow
Center for Integrative and Translational Bioimaging, University of Würzburg, Würzburg,
97070, Germany.

⁴University of Cambridge / NHS Blood and Transplant, Wellcome-MRC Cambridge Stem Cell
Institute, Jeffrey Cheah Biomedical Centre, Cambridge Biomedical Campus, University of
Cambridge, Cambridge, CB2 0AW, UK.

⁵School of Medical Sciences, University of New South Wales, Sydney, NSW 2052, Australia.

⁶Cumming School of Medicine, University of Calgary, Calgary, AB, T2N 1N4, Canada.

*** Correspondence and requests for materials should be addressed to X. Z. (email:**
**xz14926@bristol.ac.uk) or to A.W. P. (email: A.Poole@bristol.ac.uk).**

26 **Abstract**

[revised manuscript text omitted]

WHAT IS P2 THAT
 APPEARS TO BE
 THE DOMINANT
 SPECIES
 AFTER 18 PASSAGES?

SHOULD'VE
 MEASURED
 ANNEXIN V OR
 P-SELECTIN
 SURFACE
 LEVELS OR
 JON-A TO SHOW
 THAT THESE AREN'T
 ACTIVATED
 CYTOPLASMIC
 FRAGMENTS

to generate platelets (Fig. 2c). Altogether, we could generate physiological numbers of platelets
after multiple recirculation of MKs through the lung microvasculature.

**Mechanisms of platelet generation in mouse vasculature**

The *ex vivo* mouse heart-lung model (Fig.1a) can be a useful tool to allow artificial ventilation
with either ambient air or with pure nitrogen, or no ventilation, to assess the roles of physical
ventilation and gaseous oxygen in regulating MK biology and thrombogenesis. We first
explored whether air ventilation is essential for platelet generation in our model. In the absence
of ventilation, the numbers of platelets generated per MK in the perfusate still gradually
increased with increasing passages (498.4 ± 117.9 platelets/MK, Fig.2a), but the numbers
generated were substantially lower than in the air-ventilated condition. Two-photon
microscopy of fixed lung sections after 18 passages showed that fewer **generated platelets could**
**be seen** in the lung microvasculature (Fig.2b and Supplementary Movie 4), compared to the
air-ventilated lung. Therefore this indicates that air ventilation is important in platelet
generation in the lung, but may result either from an effect directly on MKs and/or through an
effect on endothelial viability. Pulmonary endothelial cells (ECs), which play key roles in gas
exchange in the lung¹⁸, interact closely with MKs as they passage through the vasculature. We
therefore compared their viabilities (determined by the Calcein Deep Red retention assay) and
mitochondrial membrane potential in the lungs under air ventilation or unventilated conditions
**for approximately 2 hours**. Surprisingly, ECs from preparations of unventilated lungs were
fully viable, and comparable with those ventilated under air (Fig.3a). However, the mean
intensity of TMRM of ECs from unventilated lungs was approximately half that of air-
ventilated lung or fresh lung (Fig.3b).

Strikingly, when lungs are ventilated with pure nitrogen to completely de-oxygenate the heart-
lung preparation, the number of generated platelets in the perfusate was almost ablated, reduced
to just 43.1 ± 16.7 platelets/MK after 18 passages (Fig.2a). Two-photon imaging of nitrogen-
ventilated lungs showed mature MKs were trapped in the lung vasculature (Fig.2b and

Supplementary Movie 5), a feature not observed under air ventilation or unventilated lung.
Importantly, the mean intensity of endothelial cell TMRM was approximately halved by
nitrogen-ventilation relative to air-ventilated controls, and equivalent to the unventilated lung

(Fig.3b). **NEED TO BE CAREFUL HERE. SHOW FSC COMPARE TO DONOR PLTS AND NEED TO MEASURE
MARKER OF ACTIVATION (EG, ANNEXIN V) TO MAKE YOUR STATEMENT OR NEED TO MODIFY STATEMENT.
ALSO IS THIS A SIZE SELECTED STUDY. NO BIG MKS DRAQ5 POSITIVE COME THROUGH?**

To further verify whether the structural arrangement of the lung capillary bed could mediate
platelet generation, we designed a polydimethylsiloxane (PDMS)-based (gas permeable)
microfluidic chamber with channel arrangement mimicking tissue microcirculation. The
channels were of uniform depth of 10 μm throughout, where the entry and exit channels had a
width of 100 μm and where branches emerged halving the channel width each time, to a
minimal width of 12.5 μm , as per the diagram shown in Fig.3c. This channel arrangement
allowed us to flow through cells and determine platelet generation in the perfusate after
repeated passage. Fig.3e shows that the numbers of generated platelets per MK, when MKs are

flowed through the microfluidic chamber conditioned in normal air, gradually increased with
increasing passages, similar to the numbers generated in the unventilated lung, with $492.3 \pm$
47.6 platelets/MK after 18 passages. The generated platelets were **live** anuclear platelets (Fig.

**WHAT DOES THIS
MEAN HERE?
SHOW DONOR PLTS
FOR COMPARIOSN**

3d). **We also conditioned microfluidic chambers with pure nitrogen to completely de-
oxygenate them, causing the generation of platelets to be almost ablated, reducing them to 56.4
± 1.4 platelets/MK after 18 passages, similar to those generated in lungs ventilated with pure**

**nitrogen** (Fig. 2b). Altogether, these data suggested that (1) air-ventilation and **healthy ECs** are
required for MKs to generate physiological levels of platelets in the heart-lung preparation; (2)
**the structural arrangement of the pulmonary microcirculation plays a role in platelet generation;**

(3) lack of ventilation or nitrogen-ventilation for 2 hours caused partial loss of the
mitochondrial membrane potential in pulmonary ECs; (4) **exclusion of oxygen from either the
lung-heart system or the microfluidic system ablates platelet generation.**

**YOU DIDN'T TEST
THIS HERE, SO
MODIFY YOUR
STATEMENT.**

180 **Generated platelets are morphologically and functionally normal**

**AGAIN NOT TESTED.
PLEASE MODERATE SENTENCE**

**DID THE MKS NOW CLOG THE
CHANNELS?**

NEED TO MAKE CLEAR THIS IS NOT MICROFLUIDIC "PLTS".

IF THE DATA IS AVAILABLE, WHAT IS THE CHANGE IN THE TWO POPULATIONS WITH RECYCLING NUMBER

We next determined whether generated platelets display classical morphology and function.
Platelets display an almost uniquely characteristic sub-plasma membrane microtubular ring,
running circumferentially in resting platelets^{19,20}. Our generated platelets, immunolabelled for
α -tubulin, display this characteristic ring structure (Fig. 4a), and the mean size of the cells is
larger than controls ($3.6 \pm 0.2 \mu\text{m}$ vs $1.9 \pm 0.1 \mu\text{m}$, Fig. 4b). However, it is also clear that there
appear to be two subpopulations of generated platelets, based on their diameter ranges as shown
in Fig. 4b: approx. 33% of generated platelets (diameter range: 1.7-2.4 μm) have sizes similar
to control platelets (diameter range: 1.2-2.4 μm) and 67% of generated platelets (diameter
ranges: 3.7-5.6 μm) are significantly larger than control platelets. We next visualized the
ultrastructure of generated platelets by transmission electron microscopy (TEM, Fig. 4c), after
depletion of host platelets using anti-GPIb α antibodies. Generated platelets displayed a discoid
shape with classical characteristics including α -granules, dense granules, mitochondria, open
canalicular system, and microtubule coils.

We then determined the functionality of generated platelets from mouse heart-lung model by
comparing against control mouse platelets. Both generated and control platelets showed
equivalent responses to agonists (CRP and thrombin) in terms of integrin $\alpha\text{IIb}\beta\text{3}$ activation and
degranulation (P-selectin expression, Fig.5a). Given that generated platelets appeared to
segregate into two size subpopulations, we then compared the responses in these two
subpopulations. The subpopulation with the larger size (diameter ranges: 3.7-5.6 μm) were
more responsive, by comparison with the subpopulation with the smaller size, to thrombin and
CRP in both integrin $\alpha\text{IIb}\beta\text{3}$ activation and P-selectin expression. It has been shown that larger
platelets are more responsive^{21,22}, and our data are therefore consistent with this observation.

We next compared the key glycoprotein expression on the surface of generated platelets. The
proportion of cells expressing CD61 and CD42b (Supplementary Fig.1g), and the mean
fluorescence intensity (MFI) of those markers (Fig.5b), were comparable between generated
and control platelets. The MFI of three collagen receptors CD42d²³, CD49b and GPVI²⁴ was

WHERE IS THESE DATA AND WHAT DO "MORE RESPONSIVE" MEAN? IDEALLY, USED AN AGONIST DOSE RESPONSIVENESS

[revised manuscript text omitted]

790

791

792
793

Fig. 1

DIFFICULT TO READ.
MAKE BLACK AND MOVE
CLOSER TO CIRCLED
AREAS?

**Fig. 1: Mouse platelets are generated from megakaryocytes passed multiple times**

**through mouse pulmonary vasculature *ex vivo*. Mouse megakaryocytes (MKs), labelled**

**with CD41-PE or CD41-FITC antibodies, were passed repeatedly through the pulmonary**

**vasculature *ex vivo*. Lungs were ventilated with air throughout (b and d-g). a Diagram**

**illustrating the approach to generating mouse platelets. End-expiratory positive pressure was**

**applied to prevent lung collapse. b Intact MKs (showing a circular shape and central nucleus)**

**were imaged from samples of perfusates after passing the indicated number of times through**

lung vasculature. Quantification was from at least 250 fields of view, counting at least 230 cells
in total and displayed as a percentage of total number of cells. Numbers above each column
indicate significant difference to other passages. $P < 0.05$ was considered statistically
significant and determined using unpaired *t*-test. Data are from at least 5 independent
experiments. **c** *In vivo* demonstration that intact mouse MKs pass through the pulmonary
vasculature. Mouse MKs were stained with CellTracker™ Red CMTPX dye (red) and Hoechst
33342 (blue) prior to injection into the right external jugular vein of an anaesthetized recipient
C57/Bl6 mouse. Blood was collected from the left common carotid artery and cells were
imaged by confocal fluorescence microscopy. Images shown are representative of at least 4
independent experiments. Scale bar: 10 μ m. **d** Gating strategy for quantification of generated
platelets. The number of generated platelets in the perfusate collected after the 18th passage
was determined by the number of CD41(+) events in gate P1. **e** Events in P1 gate (from the
experiment shown in Fig. 1d) are defined as generated platelets (indicated by the red arrow),
with higher mean fluorescence compared to those derived from control IgG-PE- treated MKs.
Gate P1 also captures CD41-negative cells, which include stem cells (despite concentration of
MKs on a 1.5%/3% BSA gradient) and host-derived platelets. The viability of the generated
platelets, and whether they contain DNA, were checked by Calcein AM and DRAQ5 dyes,
respectively. **f** Mitochondrial membrane potential in generated and control platelets was
determined by Tetramethyl rhodamine methyl ester (TMRM) accumulation in active
mitochondria and measured by FACS. **g** TMRM signals from **f** were quantified and displayed
as mean \pm S.E.M. (n=5).

Fig. 2

**Fig. 2: Quantification of platelets generated by passage of megakaryocytes through**832 **mouse pulmonary vasculature *ex vivo*. Mouse megakaryocytes (MKs), labelled with CD41-**833 **PE or CD41-FITC antibodies, were passed repeatedly through the pulmonary vasculature *ex***834 ***vivo*. Lungs were ventilated with air, pure nitrogen or without ventilation. a** The number of835 **generated platelets per megakaryocytes (MKs) present in the perfusates from different passage**836 **numbers in lungs either ventilated with air (black circles), pure nitrogen (red triangles), or**837 **without ventilation (blue squares) were measured by FACS. Data are mean and S.E.M. (n as**838 **indicated). $P < 0.05$ was considered statistically significant and determined using two-way**

ANOVA with Tukey's multiple comparisons test. **b** Stained MKs (CD41-FITC, green) were
passaged through pulmonary vasculature *ex vivo* 18 times, and lung tissue was fixed and sliced
followed by visualization of 20 stacked focal planes by two-photon microscopy. Lungs were
either ventilated with air or pure nitrogen, or were not ventilated, as indicated. Mouse lung
without MKs passed through served as control. Images shown are representative of at least 4
independent experiments. scale bar: 20 μ m. **c** Numbers of platelets generated per MK in
perfusate and retained in mouse lung under air ventilation, were calculated and displayed as
mean \pm S.E.M. (n as indicated). As a control, MKs passed through 21G needles 18 times
generated no platelets, as indicated.

Fig. 3

DON'T UNDERSTAND THE SIGNIFICANCE OF DIVIDING INTO 4 QUADRANTS AND USE OF GREEN AS WELL AS THE BELL SHAPE OF DRAQ5 WITH NO COMPARATIVE. FOR EITHER GRAPH. ALSO THE SMALL FONT AND COLORS FOR THE QUADRANT DESIGNATION IS DIFFICULT TO SEE

Fig. 3: Role of pulmonary endothelial cell health and microvascular structure on platelet generation. a-b Pulmonary endothelial cells (ECs) were isolated from perfused lungs under air- (black) or pure nitrogen-ventilation (red) or without ventilation (blue) for approximately 2 hours. ECs from fresh lung tissue served as control (gray). ECs were defined by staining with FITC-conjugated anti-CD31/PECAM-1 or anti-CD102/ICAM-2 antibodies. Data are from 5

independent experiments. $P < 0.05$ was considered statistically significant and determined
using unpaired *t*-test. **a** The viability of pulmonary ECs were determined by Calcein Deep Red
retention and displayed as mean of $\% \pm$ S.E.M. **b** The mitochondrial membrane potential was
determined by accumulation of Tetramethyl rhodamine methyl ester (TMRM) in active
mitochondria and displayed as mean fluorescence intensity \pm S.E.M. **c** Design of microfluidic
chamber mimicking a physiological pulmonary vascular system (details shown in Methods),
including a photomicrograph of the smallest channels in the system and indication of the flow
direction by arrows (red). Dimensions indicated on the figures are width of channels. **d-e**
Mouse MKs prelabelled with CD41-PE were repeatedly pumped through the microfluidic
chamber. **d** The viability of generated platelets from the microfluidic chamber was determined
by CD41 and Calcein AM staining (CD41+/Calcein AM+ in upper right quadrant, Q1-UR in
green). All generated platelets identified in this way showed no DNA content (DRAQ5 -ve
staining). **e** Quantification of generated platelets per megakaryocyte in perfusates under air
(purple circles) or pure nitrogen conditions (green circles), measured by FACS. For
comparison, numbers of platelets generated in the unventilated lung-heart system (blue
squares), from Fig. 2a, are shown. Data are mean and S.E.M. (n as indicated). $P < 0.05$ was
considered statistically significant and determined using two-way ANOVA with Tukey's
multiple comparisons test.

Fig. 4

**Fig. 4: Generated platelets demonstrate typical physical features comparable to control**

**platelets. a-b** Mouse megakaryocytes (MKs), prelabelled with CD41-FITC (green), were

**passaged repeatedly through the pulmonary vasculature *ex vivo*. Lungs were ventilated with air**

**throughout. a** Perfusates from ex vivo heart-lung preparation, containing both generated
platelets (white arrow) and host platelets (blue arrow), were stained for α -tubulin (magenta),
and confocal images shown as a mixed population in the top panels. More detailed images of
α -tubulin rings are shown in the magnified images in the middle panel (generated platelets) and
bottom panel (control platelets). Images are representative of 3 independent experiments. Scale
bars: 2 μ m. **b** The diameter of platelets (40 platelets from 3 independent experiments) from **a**
was measured using Fiji (ImageJ-Win64), **with diameters of generated platelets: $3.6 \pm 0.2 \mu$ m**
**vs control platelets: $1.9 \pm 0.1 \mu$ m. In contrast to control platelets, there were two subpopulations**
**of generated platelets based on their diameter ranges: approx. 33% of generated platelets**
**(diameter range: 1.7-2.4 μ m) have sizes similar to control platelets (diameter range: 1.2-2.4**
**μ m) and 67% of generated platelets (diameter ranges: 3.7-5.6 μ m) are significantly larger than**
**control platelets. Data are presented as mean \pm S.E.M. $P < 0.05$ was considered statistically**
**significant and determined using Mann-Whitney U test. c** Ultrastructures of generated platelets
and control platelets visualized by transmission electron microscopy. **Host platelets were**
**depleted by intraperitoneal administration of anti-GPIb α antibody R300 prior to perfusing MKs**
**through the heart-lung preparation under air-ventilation.** Subcellular structures are shown and
annotated as abbreviations, in the high magnification images: α -G, α -granules; σ -G, σ -granules
or dense bodies; Mit, mitochondria; OCS, open canalicular system; MTC, microtubule coils;
RBC, red blood cells. Scale bars: 2 μ m in the images with low magnification, 500 nm in the
images with high magnification. Images shown are representative of 5 independent
experiments.

Fig. 5

**Fig. 5: Generated platelets are functionality comparable to control platelets. Mouse**
**megakaryocytes (MKs), labelled with CD41-FITC or CD41-PE antibody or DiOC6 dye, were**
**passed through pulmonary vasculature *ex vivo* 18 times. Lungs were ventilated with air**
**throughout. a** Mean fluorescence intensity (MFI, a.u.), measured by FACS, of integrin α IIB β 3
**activation (JON/A binding, PE-conjugated) and P-selectin expression (PE-conjugated) induced**
**by 2 U/mL thrombin or 5 μg/mL CRP in washed generated platelets (CD41-FITC) compared**
**to washed control platelets. Data from generated platelets were pooled as total generated**

platelets (black dots), and also segregated by platelet size (diameter $<2.4 \mu\text{m}$ as pink dots,
diameter $3.7\text{-}5.6 \mu\text{m}$ as light brown dots), compared to control platelets (blue dots). Generated
platelets sizes were estimated using Flow Cytometry Polystyrene Particle Size Standard Kit
(Cat. PPS-6). Data were expressed as mean \pm S.E.M. (n as indicated). $P < 0.05$ was considered
statistically significant and determined using two-way ANOVA with Tukey's multiple
comparisons test. **b** Surface glycoproteins were measured by FACS. Generated platelets were
defined by staining with anti-CD41-PE antibody. Surface glycoproteins were stained with
different FITC-conjugated antibodies as indicated. Data are presented as mean \pm S.E.M. (n=6)
of FITC intensities. $P < 0.05$ was considered statistically significant and determined using
unpaired *t*-test. **c** Images of a representative platelet-rich thrombus. Generated platelets
(showing as blue in colour, stained with both DiOC6 (cyan) and CellTracker™ Red CMTPX
dye (magenta)) occupied all levels of the thrombus while host platelets (stained with
CellTracker™ Red CMTPX dye alone, magenta) were mainly situated on top of thrombus.
Images are representative of 5 independent experiments. Scale bars as indicated. **d-e** MFI
profiles of R1 and R2 (Region 1 and Region 2 from Fig. 5c) along the z-axis. **d** MFI profile of
R1 along the z-axis. In R1, generated platelets occupied the lower part of the thrombus up to
$\sim 12 \mu\text{m}$ (both cyan and magenta signals increased simultaneously), whilst beyond this point
host platelets were predominant (as magenta signals were stronger than green beyond $12 \mu\text{m}$).
**e** MFI profile of R2 along the z-axis. In R2, this part of thrombus was composed only of
generated platelets as both cyan and magenta signals changed simultaneously along z-axis.

Fig. 6

**Fig. 6: Megakaryocytes show nuclear marginalization and enucleation prior to**
 **fragmentation.** Megakaryocytes (MKs) from C57BL/6 mice, labelled with CD41-PE

antibody (red) and Hoechst 33342 (blue), were passaged repeatedly through the pulmonary
vasculature of a C57BL/6 mouse *ex vivo*. Lungs were ventilated with air throughout. **P < 0.05**
**was considered statistically significant and determined using unpaired *t*-test.** **a** Representative
images of MKs derivatives during the process of platelet generation: nuclear polarization and
enucleation, where the nucleus is marginalized, of irregular shape or in the process of ejection
from the cell; naked nuclei, where the ejected nucleus is larger than 20 μm in diameter and
**free from the parent cell and/or partially encased in thin/patchy plasma membrane; MKs**
**without nuclei**, where the MKs have an approximately circular shape but without nuclei; and
**large anuclear objects**, where ghost cells are of irregular shape and larger than 10 μm in their
longer axis. **b** Cells were imaged from samples of perfusates after passage numbers 1, 2, 3, 6
& 9 through murine lung vasculature *ex vivo*. Five subgroups of MKs and their derivatives, as
described above in **a** and in Fig. 1b, were quantified as a percentage of total number of cells.
Quantification was from at least 250 fields of view, counting at least 230 cells in total for each
of the subgroups. Data are from at least 5 independent experiments and presented as mean \pm
S.E.M. **c-e** Nuclear lobes of MKs fragment into small condensed sub-nuclei. **c** Representative
images, from n=3, of naked nuclei generated after multiple passages (as indicated) of mouse
MKs through pulmonary vasculature. Scale bars: 5 μm . **d-g** **The depth **d**, aspect ratio **e**, major**
**axis **f** and minor axis **g** of sub-nuclei decreased substantially with increasing passages. These**
**parameters were measured from after 3 passages to after 18 passages: depth 8.9 μm to 5.5 μm ,**
**aspect ratio 1.8 to 1.1, major axis 14.1 μm to 6.5 μm , minor axis 8.1 μm to 5.7 μm . Each**
**symbol represents one sub-nucleus. Numbers above each column indicate significant**
**difference to other passages.** Data are from 3 independent experiments displayed as mean \pm
S.E.M.

**Fig. 7: Tropomyosin 4 is required for the final steps in platelet generation in the**

**pulmonary vasculature.** Megakaryocytes (MKs) from Tropomyosin4^{-/-} (*Tpm4*^{-/-}) mice

labelled with FITC-conjugated anti-CD41 antibody, were passaged repeatedly through the

pulmonary vasculature of a C57BL/6 mouse *ex vivo*. For controls, wild-type (WT) MKs were

stained either with FITC-conjugated anti-CD41 or isotype antibodies. Lungs were ventilated

with air throughout. **a** Representative FACS dot plot images are shown for generated platelets

(CD41-FITC positive events are within the red square). **b** Numbers of generated platelets

1036 per *Tpm4*^{-/-} MKs, or control WT MKs, in perfusates after different passage numbers through

WT lung, were quantified by FACS. *Tpm4*^{-/-} platelets were consistently undetectable after up

to 18 passages, in the perfusate. Data shown are platelets generated per MK from either *Tpm4*^{-/-}

1039 ^{-/-} MKs or control WT MKs and displayed as mean ± S.E.M. (n as indicated). **P < 0.05 was**

1040 considered statistically significant and determined using unpaired *t*-test. **c** Cells were imaged
from samples of perfusates after passage numbers 1, 2, 3, 6 & 9 through murine lung
vasculature *ex vivo*. Cells were morphologically classified as 5 subgroups: intact MKs (as per
Fig. 1b) and MK derivatives (shown in Fig. 6a) and quantified as a percentage of total number
of cells. Quantification was from at least 150 fields of view, counting at least 170 cells in total
for each of the subgroups. Numbers above each column indicate significant difference to other
passages within each group. Dollar sign (\$) above columns represents significant difference to
corresponding wild-type column in Fig.6b. **P < 0.05 was considered statistically significant and**
**determined using unpaired *t*-test.** Data are from 4 independent experiments and displayed as
mean ± S.E.M. **d** Abundant fluorescent objects, ~10 μm diameter, were visible in sections of
mouse lung after 18 passages of stained *Tpm4*^{-/-} MKs, as shown in extended focus stacks of 20
continuous two-photon planes of lung. Images shown are representative of 3 independent
experiments. Scale bar: 20 μm.

Fig. 8

**Fig. 8: Intravital two-photon microscopy of bone marrow megakaryocytes in live mouse**1070 **calvarium.** Bone marrow vasculature was visualized by intravenous injection of anti-CD105-

AlexaFluor 546 antibody and AlexaFluor 546-labeled BSA (red). Megakaryocytes (MKs) and

**their derivatives** were stained intravenously with anti-GPIX-AlexaFluor 488 antibody (cyan).1073 Both wild-type (WT) and *Tropomyosin 4*^{-/-} (*Tpm4*^{-/-}) MKs were generally seen in close contact1074 with the bone marrow sinusoidal walls. **a** Representative images of large fragments of WT and1075 *Tpm4*^{-/-} MKs (white arrows) within sinusoidal vessels releasing heterogeneous structures in1076 the direction of blood flow, at time points indicated. **b** Representative images of WT and *Tpm4*^{-/-}

MKs, with cell bodies within the marrow space, producing extensions into sinusoids (yellow

arrowheads). **c** Representative images of WT and *Tpm4*^{-/-} MKs within the sinusoid, showing1079 cellular extensions (white circles). Images were taken from 6 WT and 6 *Tpm4*^{-/-} mice. Scale1080 bars, 50 μ m.

.

Fig. 9

Fig.9: Schematic diagram of the steps in platelet generation from mature

megakaryocytes. Diagram showing platelet generation pathway from intact mature

megakaryocytes (MKs) to final platelet formation, by repeated passage of MKs through the

pulmonary vasculature. The process involved nuclear polarization, enucleation, gradual

cytoplasmic fragmentation into platelets and nuclear fragmentation and condensation.

a Representative images of MKs derivatives

a Large fragments of MKs releasing heterogeneous structures within sinusoids

b MKs producing extensions into sinusoids

c MKs within sinusoid showing cellular extensions

REVIEWER COMMENTS

Reviewer #1 (Remarks to the Author):

I thank the authors who responded satisfactorily to all my comments and the manuscript is acceptable for publication

Reviewer #2 (Remarks to the Author):

Thank you for the responses. The details of the heart-lung prep are now much clearer. I have one remaining question. Did the application of the end-expiration device (Gottlieb valve) affect any of the results such as those in Figure 1b or Figure 2a? If possible, data should be shown with and without the device.

Thanks for this question. Here is the comparison with or without Gottlieb valve (below). There is no significant difference between these two conditions in terms of platelet generation in the perfusates over passages.

We have not included this data in our text, because the data in that Figure showed no significant difference with or without Gottlieb valve.

Reviewer #3 (Remarks to the Author):

WHERE'S SUPPL FIG.1A IN TEXT? CHANGE ORDER IN FIG?

Thanks for the good suggestions. We now have modified these and highlighted in red colour in the text.

nitrogen-ventilation relative to air-ventilated controls, and equivalent to the unventilated lung
 (Fig.3b). NEED TO BE CAREFUL HERE. SHOW FSC COMPARE TO DONOR PLTS AND NEED TO MEASURE
 MARKER OF ACTIVATION (EG, ANNEXIN V) TO MAKE YOUR STATEMENT OR NEED TO MODIFY STATEMENT.
 ALSO IS THIS A SIZE SELECTED STUDY. NO BIG MKS DRAQ5 POSITIVE COME THROUGH?
 To further verify whether the structural arrangement of the lung capillary bed could mediate
 platelet generation, we designed a polydimethylsiloxane (PDMS)-based (gas permeable)
 microfluidic chamber with channel arrangement mimicking tissue microcirculation. The
 channels were of uniform depth of 10 μm throughout, where the entry and exit channels had a
 width of 100 μm and where branches emerged halving the channel width each time, to a
 minimal width of 12.5 μm , as per the diagram shown in Fig.3c. This channel arrangement
 allowed us to flow through cells and determine platelet generation in the perfusate after
 repeated passage. Fig.3e shows that the numbers of generated platelets per MK, when MKs are
 flowed through the microfluidic chamber conditioned in normal air, gradually increased with
 increasing passages, similar to the numbers generated in the unventilated lung, with $492.3 \pm$
 47.6 platelets/MK after 18 passages. The generated platelets were live anuclear platelets (Fig.
 3d). WHAT DOES THIS MEAN HERE?
 SHOW DONOR PLTS FOR COMPARIOSN
 We also conditioned microfluidic chambers with pure nitrogen to completely de-

In Fig. 1d, Gating strategy for quantification of generated platelets. The left panel is the donor (control) platelets.

WHAT DOES THIS MEAN HERE? SHOW DONOR PLTS FOR COMPARIOSN

In Fig. 1e, we showed the Calcein AM staining to determine the viability for generated platelets-“**live**”

We now have added donor platelets (control platelets) stained with Calcein AM and Draq5 in Fig. 1 e. Supplementary Fig 1e shows DRAQ5 staining for MKs, which provides a positive control for this stain.

1. In Fig 1d, why is that P1 window more “plt-like” at 0 passages and the P2 larger after 18 passages. Seems the reverse of what to expect.

- 1) MKs were enriched using a **1.5%/3% BSA gradient** (4 mL 1.5% BSA + 4 mL 3% BSA) for 1 hour, before infusing into the lung-heart system or microfluidic chambers. This step removes most small cells or particles.
- 2) In the P1 window, at 0 passages, in our MKs preparation, only **1.3-7.6% events** were **CD41+**. That is **8.2±2.0 CD41+ events per MK** in MKs suspension (N=4), as shown in **Fig 2c-pre-needle**. We have not identified these CD41- events.
- 3) The numbers of generated platelets in the perfusates over different passages are all given after subtraction of the number of CD41+ events at 0 passage.
- 4) We have now added new **Supplementary Fig 1e** showing the staining of events in the P2 gate after 18 passages, including Draq5 staining, which showed Draq5 staining was efficient.

WHAT IS P2 THAT APPEARS TO BE THE DOMINANT SPECIES AFTER 18 PASSAGES?

The events in P2 window after 18 passage predominantly are white blood cells, red blood cells and some CD41+ events, as shown in Supple Fig. 1e----events in P2 from Fig. 1d. 10 ul of MK suspension or perfusates were measured by FACS.

We now have added “whole blood” from host mouse, as a comparison, as gating strategies in Figure 1d.

2. In Fig 1f, annexin v or p-selectin or Jon A binding would have been useful and more commonly done to show that you were generating “happy” platelets.

SHOULD'VE MEASURED ANNEXIN V OR P-SELECTIN SURFACE LEVELS OR JON-A TO SHOW THAT THESE AREN'T ACTIVATED CYTOPLASMIC FRAGMENTS

The results of P-selectin expression and integrin activation (JON/A binding assay) for generated platelets in lung-heart system in response to agonists (thrombin and CRP) are shown in Fig. 5a and described under “**Generated platelets are morphologically and functionally normal**” section in **Result part**.

Fig. 4a shows that generated platelets display an almost uniquely characteristic sub-plasma membrane **microtubular ring**, running circumferentially in resting platelets. Fig. 4c showed the **microtubule coils**. These features **make us believe that they are platelets rather than cytoplasmic fragments**.

equivalent responses to agonists (CRP and thrombin) in terms of integrin α Ib β 3 activation and
degranulation (P-selectin expression, Fig.5a). Given that generated platelets appeared to
segregate into two size subpopulations, we then compared the responses in these two
subpopulations. The subpopulation with the larger size (diameter ranges: 3.7-5.6 μ m) were
more responsive, by comparison with the subpopulation with the smaller size, to thrombin and
CRP in both integrin α Ib β 3 activation and P-selectin expression. It has been shown that larger
platelets are more responsive^{21,22}, and our data are therefore consistent with this observation.
We next compared the key glycoprotein expression on the surface of generated platelets. The

WHERE IS THESE DATA AND WHAT DO "MORE RESPONSIVE MEAN? IDEALLY, USED AN AGONIST DOSE RESPONSIVENESS

In the thrombus formation assay, shown in **Fig. 5c-e and Supplementary Movie 6**, generated platelets occupied all levels of the thrombus whilst control platelets were mainly situated on top of the thrombus. This suggested that **generated platelets had a higher responsiveness to collagen, or were primary reactors to it**.

oxygenation level, causing the generation of platelets to be almost ablated, resulting in a count
± 1.4 platelets/MK after 18 passages, similar to those generated in lungs ventilated with pure
nitrogen (Fig. 2b). Altogether, these data suggested that (1) air-ventilation and healthy ECs are
required for MKs to generate physiological levels of platelets in the heart-lung preparation; (2)
the structural arrangement of the pulmonary microcirculation plays a role in platelet generation;
(3) lack of ventilation or nitrogen-ventilation for 2 hours caused partial loss of the
mitochondrial membrane potential in pulmonary ECs; (4) exclusion of oxygen from either the
lung-heart system or the microfluidic system ablates platelet generation.

YOU DIDN'T TEST THIS HERE, SO MODIFY YOUR STATEMENT.

In our manuscript, we have defined “healthy ECs” as “ECs with normal mitochondrial membrane potential”.

3. While EC injury/death by inhalation of nitrogen or no ventilation is mentioned, it needs to be **given more equal time** as a possible explanation in the Results and Discussion for why platelet formation was not seen.

We agree, and we did actually use similar time exposures (around 2 hours) for all conditions, including no ventilation and ventilation with nitrogen. This is described in the ‘Flow Cytometry’ section in Methods, and we have highlighted this in red in the text.

In the absence of ventilation, the numbers of platelets generated per MK in the perfusate still gradually increased with increasing passages (498.4 ± 117.9 platelets/MK, Fig.2a), but the numbers generated were substantially lower than in the air-ventilated condition.

When lungs are ventilated with pure nitrogen, the number of generated platelets in the perfusate was almost ablated, reduced to just 43.1 ± 16.7 platelets/MK after 18 passages (Fig.2a), due to mature MKs were trapped in the lung vasculature (Fig.2b and Supplementary Movie 5).

Staining for surface markers compatible with **endothelial injury** like loss of surface thrombomodulin or extrusion of VWF would have been important.

- 1) Here we investigated the **general ‘health’ state of the endothelium** by assessing the resting membrane potential across mitochondrial membranes, using a standard dye for this, TMRM. If cells are healthy and have functioning mitochondria, the signal will be bright, and we have used TMRM staining as a measure of endothelial cell health. We have therefore now modified our text, generally to indicate endothelial with normal mitochondria.
- 2) An in vitro study has shown that the increase in thrombomodulin **was closely correlated with** the loss of cell viability (2002 Nov; 107(3): 340–349). Our ECs from air-ventilated-, pure nitrogen-ventilated or unventilated lung all show normal viability, as measured by retention of Calcein Deep Red.

NEED TO MAKE CLEAR THIS IS NOT MICROFLUIDIC "PLTS".

IF THE DATA IS AVAILABLE, WHAT IS THE CHANGE IN THE TWO POPULATIONS WITH RECYCLING NUMBER

We next determined whether generated platelets display classical morphology and function.

Platelets display an almost uniquely characteristic sub-plasma membrane microtubular ring,

running circumferentially in resting platelets^{19,20}. Our generated platelets, immunolabelled for

α -tubulin, display this characteristic ring structure (Fig. 4a), and the mean size of the cells is

larger than controls ($3.6 \pm 0.2 \mu\text{m}$ vs $1.9 \pm 0.1 \mu\text{m}$, Fig. 4b). However, it is also clear that there

appear to be two subpopulations of generated platelets, based on their diameter ranges as shown

in Fig. 4b: approx. 33% of generated platelets (diameter range: $1.7\text{-}2.4 \mu\text{m}$) have sizes similar

to control platelets (diameter range: $1.2\text{-}2.4 \mu\text{m}$) and 67% of generated platelets (diameter

ranges: $3.7\text{-}5.6 \mu\text{m}$) are significantly larger than control platelets. We next visualized the

This is good to clarify in the text, and we now have added in the 'heart-lung system' or 'microfluidic chamber' to define the generated platelets from either approach specifically.

In regard to the question about 2 sub-populations of generated platelets, we image the cells after 18 passages, and assess using ImageJ. We have no data about the dynamic changes in these 2 sub-populations of platelets generated over different passages.

The microfluidic studies are complementary but not as well developed.

1. The similarity between the microfluidic design and lung vasculature is not demonstrated and the **discussion** should be altered to reflect that especially as no other design was studied.

Thanks for your suggestion and we now have changed the word "**mimic**" into "**simulate**" in our text.

2. The device has no endothelial lining and that limitation and its implications discussed.

In this manuscript, we have used the microfluidic design to simulate a series of parallel microvascular channels, or network of these, to demonstrate the importance of these small microchannels in platelet formation, and also to assess or exclude the role of endothelial cells in this, as these are absent from these channels.

We agree that the current microfluidic design has a lot of potential for further development, including different patterning and inclusion of endothelium as the reviewer has indicated, and this is something that we are actively pursuing particularly for generating larger scale **bioreactors** for platelet generation in vitro. We have now added a sentence to the Discussion section to include this point about endothelial cell lining in the microfluidic chamber.

3 Fig 3 was shown with no mouse platelet control and that would have been an important comparative. Again, markers of activated platelets should have been measured.

NEED TO BE CAREFUL HERE. SHOW FSC COMPARE TO DONOR PLTS AND NEED TO MEASURE MARKER OF ACTIVATION (EG, ANNEXIN V) TO MAKE YOUR STATEMENT OR NEED TO MODIFY STATEMENT. ALSO IS THIS A SIZE SELECTED STUDY. NO BIG MKS DRAQ5 POSITIVE COME THROUGH?

Thank you for your suggestion. We have now added control platelets in Fig. 1d and e. Supple Fig. 1e showed the Draq5 staining for MKs. These are the control comparators for Fig. 1, but also for Fig. 3.

Markers of activated platelets generated from microfluidic now have been shown in Fig. 3f.

DID THE MKS NOW CLOG THE CHANNELS?

In regard to the question about our microfluidic chambers and whether MKs clog these channels when under nitrogen, we are currently establishing the approach to be able to image these dynamically in the presence of 100% nitrogen. For this reason, we have not yet been able to observe this, but can say that the channels are unlikely to be fully clogged, as suggested, since fluid does pass through the chambers in the presence of nitrogen, but that the cells have been left behind in the chamber. The fluid that exits the chamber, under 100% nitrogen, is fully clear, and devoid of cells. Therefore our understanding is that there is not fully blockade of the channels, i.e. it is not fully clogged, but clearly the cells are not able to passage through the channels.

Inflation or deflation?

This should be “inflation”, because as the transpulmonary pressure increases, narrowing of the capillaries occurs. This is described in Ref 35.

flow through the lung, and therefore the majority of MKs are likely to passage through the
capillary bed of the lung. NEED TO BE FAIR & BALANCED AND THE OXYGEN RELATIONSHIP MAY BE
PHYSIOLOGIC AND MK AND/OR EC- DEPENDENT OR ARETIFICIAL DUE TO
ENDOTHELIAL INJURY DUE TO HYPOXIA AND MK ADHERENCE TO THE
INJURED EC AND FURTHER STUDIES ARE NEEDED TO DISTINGUISH THESE.
The ability of MKs to pass through the lung microvasculature is oxygen-dependent in our ex
vivo system, as longer term exposure of the lung to pure nitrogen, to completely de-oxygenate
the lung over 2 hours, effectively caused MKs to be retained in the lung vasculature. Oxygen-
dependent MK motility might partially explain why pulmonary MK levels observed at autopsy
are increased in COVID-19 patients who had died with acute lung damage⁴³.

Oxygen, or lack of it, might affect both ECs and/or MKs, thereby impairing the mobility of MKs to pass through the lung microvasculature. We wanted to indicate that either or both is possible.

We suspect that lack of oxygen might predominantly affect MKs, because although the EC viability and mitochondria membrane potential are similarly affected in unventilated lungs compared to N₂-ventilated lungs, they differ in the number of platelets generated and number of MKs trapped in the lung vasculature. In particular, there are no MKs trapped in our unventilated model, whereas in the presence of 100% N₂, there are abundant MKs trapped. We are working on this, including growing ECs in the microfluidic system, and under different O₂ concentrations, to confirm whether O₂ predominantly affects the endothelial cells or the MKs, to affect the platelet generation process. We have therefore included the following sentence in the Discussion section: ‘Lack of oxygen may affect the platelet generation process through effects on the pulmonary vascular endothelium, or directly on

MKs, or some interaction between both cell types’.

346 fourth, multiple passage of MKs through the pulmonary vasculature in the *ex vivo* lung model
347 is crucial to induce a reproducible sequence of events (Fig. 9). These include enucleation,
348 nuclear fragmentation and condensation, prior to efficient platelet generation. This might
349 suggest an essential and dynamic conversation between the pulmonary microvasculature and

In our observations we found that nuclear fragmentation happened earlier than nuclear condensation, because we first saw the nuclear lobes were partially fragmenting from a naked MK nucleus after 3 passages (Fig. 6c and Supplementary Movie 3) and the sub nuclei became smaller and more condensed at later passages (6 and 18), as shown in Fig. 6c-f.

marrow space, in close contact with sinusoidal walls; (ii) some MKs within the marrow space
produced extensions into sinusoids; (iii) some MKs were clearly visible wholly within the
sinusoid vessels themselves and had cellular extensions; (iv) some appeared as large fragments
in the sinusoid, releasing heterogeneous structures in the direction of blood flow (Fig. 8 and
Supplementary Movies 13-14). These observations suggested that TPM4 in MKs does not play
an essential role in platelet release in the bone marrow, in contrast to its role in these cells in
the lung vasculature.

**Discussion**

In this study, we established an *ex vivo* pulmonary vascular model, which we show generates

I'D SUGGEST MODIFYING THIS. ITS EITHER THAT MK OR LARGE CYTOPLASMIC FRAGMENT RELEASE IS NORMAL IN THE MARROW. PLT FORMATION MAY BE TPM4-DEPENDENT BUT NOT THESE EARLIER PROCESSES THAT CAN GO ON IN BOTH THE MARROW AND LUNGS BUT FINAL PLT FORMATION MAY BE IMPAIRED.

Thank you very much for this discussion. We now have modified this statement and highlighted in red colour in our text. We made a figure (below) to explain this but haven't included this in our manuscript.

lung system, despite undetectable platelets in the perfusate (Fig.7a-b), abundant anuclear
 fluorescent objects, sized $\sim 10 \mu\text{m}$, could be seen in the lung vasculature (Fig.7d and
 Supplementay Movie 12). This suggests that TPM4 is required for the final steps in platelet
 generation. Importantly, our data from intravital observation of *Tpm4^{-/-}* bone marrow suggests
 that MK protrusion/extrusion process, which may be proplatelet formation in vivo, is similar
 to normal (Fig.8 and Supplementary Movies 13-14). This therefore makes it likely that the
 lower platelet count seen in *Tpm4^{-/-}* mice is a product of defective platelet formation outside of
 the bone marrow. in the lung vasculature.

SEE PRIOR COMMENT
 THAT THIS MAY
 NOT BE PROPLT
 BUT LARGE
 FRAGMENT FORMATION
 ONLY. IT MAY BE
 THAT TPM4 IS
 NOT NEEDED FOR MEGS
 TO MIGRATE OUT
 OF THE MARROW
 BUT IMPORTANT
 FOR FUTURE
 PROCESSING

These have been addressed and highlighted in red colour in the text.

DIFFICULT TO READ.
 MAKE BLACK AND MOVE
 CLOSER TO CIRCLED
 AREAS?

Thank you for this point. This has been modified in Fig. 1d.

The quadrants define the live CD41+ events, which are generated platelets. These quadrants were defined using control platelets. We have included text in the figure legend to indicate that Q3 has been defined as generated platelets, using parameters measured using control platelets. We have coloured these events as green, just to clearly delineate them from events in the other quadrants.

Regarding DRAQ5 staining, Figure 1e had already shown equivalent data for control platelets, and Supp Fig 1e shows DRAQ5 staining for MKs, which provides a positive control for this stain.

We agree that the font for the quadrant labelling was difficult to read, and so we have replaced this with new type, and re-named the quadrants in a clearer way, as quadrants Q1-4. Appropriate reference to these quadrants in the text and legend have now been amended also.

REVIEWERS' COMMENTS

Reviewer #2 (Remarks to the Author):

Thank for the clarification.

Reviewer #3 (Remarks to the Author):

No further studies or modifications needed.